# FS-KAN: Permutation Equivariant Kolmogorov-Arnold Networks via Function Sharing

**Ran Elbaz[1], Guy Bar-Shalom[1], Yam Eitan[1], Fabrizio Frasca[1], Haggai Maron[1,2]**
[1] Technion – Israel Institute of Technology
[2] NVIDIA Research

## ABSTRACT

Permutation equivariant neural networks employing parameter-sharing schemes have emerged as powerful models for leveraging a wide range of data symmetries, significantly enhancing the generalization and computational efficiency of the resulting models. Recently, Kolmogorov-Arnold Networks (KANs) have demonstrated promise through their improved interpretability and expressivity compared to traditional architectures based on MLPs. While equivariant KANs have been explored in recent literature for a few specific data types, a principled framework for applying them to data with permutation symmetries in a general context remains absent. This paper introduces Function Sharing KAN (FS-KAN), a principled approach to constructing equivariant and invariant KA layers for arbitrary permutation symmetry groups, unifying and significantly extending previous work in this domain. We derive the basic construction of these FS-KAN layers by generalizing parameter-sharing schemes to the Kolmogorov-Arnold setup and provide a theoretical analysis demonstrating that FS-KANs have the same expressive power as networks that use standard parameter-sharing layers, allowing us to transfer well-known and important expressivity results from parameter-sharing networks to FS-KANs. Empirical evaluations on multiple data types and symmetry groups show that FS-KANs exhibit superior data efficiency compared to standard parameter-sharing layers, by a wide margin in certain cases, while preserving the interpretability and adaptability of KANs, making them an excellent architecture choice in low-data regimes.

## 1 INTRODUCTION

Designing neural network architectures that respect symmetries has become a central theme in modern machine learning, with numerous works showing that leveraging symmetries can lead to improved generalization and computational efficiency (Cohen & Welling, 2016; Ravanbakhsh et al., 2017; Zaheer et al., 2017; Esteves et al., 2018; Kondor & Trivedi, 2018; Bronstein et al., 2021; Maron et al., 2019b). Equivariant models are designed so that certain transformations applied to the input correspond to predictable transformation of the output. Permutation symmetries, subgroups of the symmetric group $S_n$ that act on a vector by permuting coordinates, are an important and natural family of symmetries that appear in real-world data. Although several methods exist, the most common, general, and scalable approach for designing equivariant architectures for permutation symmetries is to encode equivariance at each layer through a group-specific parameter-sharing scheme (Wood & Shawe-Taylor, 1996). These ideas have been applied across a wide range of domains like images (LeCun et al., 1998), graphs (Maron et al., 2019b; Pan & Kondor, 2022), sets (Zaheer et al., 2017), and many more.

Kolmogorov–Arnold Networks (KANs) have recently been proposed as an alternative to traditional Multilayer Perceptrons (MLPs), replacing scalar weights with learnable univariate functions (Liu et al., 2024). This design is inspired by the Kolmogorov–Arnold representation theorem (Tikhomirov, 1991) and leads to models with potentially improved expressivity relative to MLPs, particularly when the number of parameters is constrained. Some works have recently integrated KANs into equivariant models for graphs (Bresson et al., 2024; Kiamari et al., 2024), sets (Kashefi,

2024), and images (Bodner et al., 2024). In particular, variants like Graph-KAN (Kiamari et al., 2024) have achieved competitive or superior performance on graph-structured data. Very recently, Hu et al. (2025) proposed an approach for equivariant KANs, focusing on continuous groups.

While these approaches illustrate the potential of KANs for learning data with symmetries, the general principles for constructing equivariant KA layers for the important case of arbitrary *permutation symmetry groups* remain unexplored. In particular, equivariant KANs have not yet been developed for numerous important data types that exhibit permutation symmetries, including multi-set interactions (Hartford et al., 2018), sets with symmetric elements (Maron et al., 2020), weight spaces (Navon et al., 2023; Zhou et al., 2023), hierarchical structures (Wang et al., 2020), and high order relational data (Maron et al., 2019b).

**Our approach.** In this work, we introduce Function Sharing KAN (FS-KAN), a general framework for constructing KANs with equivariant and invariant KA layers with respect to *arbitrary permutation symmetry* groups. Our construction generalizes the well-known parameter-sharing schemes used by previous equivariant architectures (Wood & Shawe-Taylor, 1996; Ravanbakhsh et al., 2017; Maron et al., 2019b) by constraining KA layers to share functions rather than weights or parameters. In particular, FS-KANs generalize several previously proposed equivariant KANs for sets (Kashefi, 2024) and images (Bodner et al., 2024). We discuss several typical function sharing scenarios that arise in natural data, like direct product symmetries and symmetries involving high-order tensors, and propose a more efficient FS-KAN variant to reduce computational and memory costs.

From a theoretical perspective, we prove that FS-KAN architectures have equivalent expressive power to standard parameter-sharing MLPs in the uniform function approximation sense, implying that both architecture classes can represent the same functions. This equivalence establishes both fundamental limitations and guarantees for FS-KAN expressivity derived from equivalent results for parameter-sharing neural networks, including expressivity analysis for parameter-sharing-based GNNs (Chen et al., 2019; Geerts, 2020; Maron et al., 2019b;a; Azizian & Lelarge, 2020) and universality theorems for invariant or equivariant networks (Zaheer et al., 2017; Yarotsky, 2022; Segol & Lipman, 2019; Maron et al., 2019c; Keriven & Peyré, 2019) — all of which directly transfer using our result to their FS-KAN counterparts.

Our experiments across multiple data types with varying types of permutation symmetries demonstrate that FS-KANs excel when learning on data with symmetries, achieving excellent parameter efficiency and significantly outperforming standard parameter-sharing networks in low-data regimes. Moreover, we show that FS-KANs inherit the enhanced interpretability and adaptability of KANs, providing both transparent feature learning and robust performance in continual learning scenarios.

**Contributions.** **(i)** We propose FS-KAN, a principled framework for designing equivariant and invariant KA layers under arbitrary permutation symmetry groups using function sharing. **(ii)** We provide a theoretical analysis showing that FS-KANs match the expressivity of parameter-sharing MLPs and use this result to transfer well-known expressivity results to FS-KANs. **(iii)** We demonstrate empirically, across multiple domains, that FS-KANs excel when training data is limited, outperforming parameter-sharing MLPs and other natural baselines in data-scarce settings.

## 2 RELATED WORK

**Equivariance in deep learning.** A function is equivariant to a group action if a group transformation of the input leads to a corresponding transformation of the output. Perhaps the most well-known example of this principle is convolutional neural networks (CNNs) (LeCun et al., 1998) that employ translation equivariant layers; more recent works have generalized equivariance to broader symmetry groups (Cohen & Welling, 2016; Ravanbakhsh et al., 2017; Qi et al., 2017; Zaheer et al., 2017; Esteves et al., 2018; Kondor et al., 2018; Weiler & Cesa, 2019; Pan & Kondor, 2022; Hands et al., 2024; Zhang et al., 2024). For instance, Zaheer et al. (2017) characterized permutation equivariant layers in the context of set inputs, paving the way for many set-based architectures such as Hartford et al. (2018); Maron et al. (2020). Kondor et al. (2018); Maron et al. (2019b;a); Pan & Kondor (2022) studied equivariant layers for graphs, and Cohen & Welling (2016) introduced a framework for building networks equivariant to additional transformations like rotations and reflections. Recent work by Navon et al. (2023); Zhou et al. (2023) leverages symmetries within neural networks to develop effective model editing and analysis techniques. As constraining models to be equivariant might lead to models with limited expressive power (Xu et al., 2019; Morris et al., 2019; Geerts, 2020), numerous works in recent years studied and characterized the expressive power of equivari-

ant models (Zaheer et al., 2017; Maron et al., 2019a;c; Dym & Maron, 2020; Ravanbakhsh, 2020; Azizian & Lelarge, 2020).

**Kolmogorov–Arnold Networks (KANs) and equivariant KANs.** KANs (Liu et al., 2024) are a recently proposed class of neural architectures motivated by the Kolmogorov–Arnold representation theorem (Tikhomirov, 1991). They provide a promising alternative to traditional MLPs due to their improved interpretability, adaptability, and parameter efficiency (Alter et al., 2024; Samadi et al., 2024; Somvanshi et al., 2024) and have demonstrated impressive performance (Xu et al., 2024; Moradi et al., 2024; Yang et al., 2024). Recent efforts to combine the strengths of KANs with the benefits of symmetry-aware learning have led to several equivariant adaptations of KANs. On graph-structured data, Bresson et al. (2024) integrates KAN into message-passing frameworks and achieves competitive performance with respect to MLP-based GNNs. Other works have introduced variants such as Convolutional KANs (Bodner et al., 2024), Graph-KAN (Kiamari et al., 2024), PointNet-KAN (Kashefi, 2024). Very recently, Hu et al. (2025) proposed equivariant KANs for continuous groups. While theoretically applicable to permutation groups, their approach requires solving numerically for equivariant layers compared to our straightforward efficient sharing schemes and, unlike our approach, cannot handle variable-sized inputs (e.g., sets and graphs). To summarize, while prior works have developed effective equivariant KANs for specific permutation groups, we provide a framework that unifies many of these works and establishes their theoretical foundations.

## 3 INVARIANT AND EQUIVARIANT FUNCTION SHARING NETWORKS

In this section, we formulate invariant and equivariant KA layers for symmetry groups $G \leq S_n$ (section 3.2). We then introduce a variant called *efficient FS-KA layer* , which can reduce both time and memory complexity. We provide concrete examples of FS-KA layers and demonstrate how our framework handles specific, yet important types of permutation groups including direct-product symmetries and high-order tensor symmetries (section 3.3). We use the notation $[n]$ to denote $\{1, \ldots, n\}$.

### 3.1 PRELIMINARIES

**Kolmogorov-Arnold Networks.** A KA layer $\Phi : \mathbb{R}^{n_{\text{in}}} \to \mathbb{R}^{n_{\text{out}}}$ has the form of $\Phi(\mathbf{x})_q = \sum_{p=1}^{n_{\text{in}}} \phi_{q,p}(x_p)$, $q \in [n_{\text{out}}]$, where $\phi_{q,p} : \mathbb{R} \to \mathbb{R}$ are learned univariate functions. While the original KAN work uses splines to parameterize these functions, alternative parameterizations have also been explored (Li, 2024; SS et al., 2024). Our construction in this section does not assume a specific parameterization. Similar to Liu et al. (2024), we will use a $n_{\text{out}} \times n_{\text{in}}$ matrix of functions to represent the layers:

$$\Phi(\mathbf{x}) = \begin{bmatrix} \phi_{1,1}(\cdot) & \ldots & \phi_{1,n_{\text{in}}}(\cdot) \\ \vdots & \ddots & \vdots \\ \phi_{n_{\text{out}},1}(\cdot) & \ldots & \phi_{n_{\text{out}},n_{\text{in}}}(\cdot) \end{bmatrix} \star \begin{bmatrix} x_1 \\ \vdots \\ x_{n_{\text{in}}} \end{bmatrix}. \tag{1}$$

Where the '$\star$' operator denotes applying the KA layer to the vector.

**Invariance, equivariance, and standard parameter-sharing.** Let $G \leq S_n$ be a group with representations $(V, \rho), (V', \rho')$. A function $L : V \to V'$ is $G$-equivariant if it commutes with the group action, i.e., for all $g \in G$ and $\mathbf{v} \in V$, $L(\rho(g)v) = \rho'(g)L(v)$. When using linear layers $L(v) = Wv$, this equivariance constraint imposes structure on the weight matrix $W$: $\rho'(g)^{-1} W \rho(g) = W$, $\forall g \in G$ (Wood & Shawe-Taylor, 1996; Ravanbakhsh et al., 2017; Maron et al., 2019b; Finzi et al., 2021) . Specifically, when $V = V' = \mathbb{R}^n$ and $\rho = \rho'$ are the permutation representations of $G \leq S_n$, the weights $W \in \mathbb{R}^{n \times n}$ must satisfy the parameter-sharing condition to be equivariant: $W_{i,j} = W_{\sigma(i),\sigma(j)}$, $\forall \sigma \in G$. For example, convolutions, which are represented as circulant matrices, follow a parameter-sharing scheme for the cyclic group $C_n$. In the case of an invariant layer $L : \mathbb{R}^n \to \mathbb{R}$, where the group acts trivially on the output space, invariance induces the following sharing scheme on the weight vector $\mathbf{w} \in \mathbb{R}^n$: $\mathbf{w}_j = \mathbf{w}_{\sigma(j)}, \forall \sigma \in G$. These constraints also generalize to higher-order vector spaces by the Kronecker product (Maron et al., 2019b). These fundamental constraints form the basis of parameter-sharing in equivariant networks, as it defines a principled approach for constructing equivariant linear *layers*. Equivariant and invariant *networks* can then be constructed using a composition of parameter-sharing linear layers and point-wise non-linearities. Going forward, we simplify the notation of the permutation group action by omitting $\rho$, formally $(\rho(g)\mathbf{x})_i = (g \cdot \mathbf{x})_i = x_{g^{-1}(i)}$, $g \in G$.

## 3.2 Invariant and equivariant Function Sharing KA layers

First, we define the Function Sharing (FS) scheme in the equivariant setting. Intuitively, and motivated by standard parameter-sharing schemes, the idea is that the functions in the layer are tied together according to the group action.

**Proposition 1.** *Let $\Phi$ be a KA layer with $n_{in} = n_{out} = n$. We say that $\Phi$ is a **G-equivariant Function Sharing (FS)** KA layer if and only if*

$$\phi_{q,p} = \phi_{\sigma(q),\sigma(p)}, \quad \forall p, q \in [n], \ \forall \sigma \in G. \tag{2}$$

*Moreover, any such G-equivariant FS-KA layer is G-equivariant (proof in Appx. B.2).*

This structure is conceptually similar to standard parameter-sharing: as illustrated in figure 1, in their matrix form, linear convolution layers and KA-convolution layers (Bodner et al., 2024) have the structure of a circulant matrix. However, unlike the linear parameter-sharing case, we note that there exist KA equivariant layers that *do not* follow FS structure. For example, consider the following $S_2$ equivariant KA layers :

$$\Phi() = \begin{bmatrix} cos(\cdot) + 2 & sin(\cdot) - 2 \\ sin(\cdot) + 3 & cos(\cdot) - 3 \end{bmatrix} \quad , \quad \hat{\Phi}() = \begin{bmatrix} cos(\cdot) & sin(\cdot) \\ sin(\cdot) & cos(\cdot) \end{bmatrix}. \tag{3}$$

Both layers compute the same function; however, only $\hat{\Phi}$ is an FS-KA layer. As we will show next, every $G$-equivariant KA layer can nonetheless be represented by an FS layer: (proof on Appx. B.3)

**Proposition 2.** *Let $\Phi$ be a G-equivariant KA layer. Then, there exists a G-equivariant FS-KA layer $\hat{\Phi}$ such that $\hat{\Phi}(\mathbf{x}) = \Phi(\mathbf{x}), \ \forall \mathbf{x} \in \mathbb{R}^n$ .*

Importantly, this result allows us to restrict attention to FS layers when designing equivariant KANs without losing generality, simplifying both theoretical analysis and practical implementation.

**Invariant layers.** We define $G$-invariant FS-KA layers as follows: (proof on Appx. B.4)

**Proposition 3.** *Let $\Phi$ be a KA layer with $n_{in} = n, n_{out} = 1$. If , $\forall p \in [n], \ \forall \sigma \in G, \ \phi_p = \phi_{\sigma(p)}$, then $\Phi$ is an G-invariant layer, and we call it G-**invariant FS KA** layer.*

Similarly, the $G$-invariant FS-KA layers can express any $G$-invariant KA layer: (proof on Appx. B.5)

**Proposition 4.** *every G-invariant KA layer has an equivalent G-invariant FS-KA layer representation.*

**Feature channels.** So far, we have considered inputs with a single feature per element. When learning on data with symmetries, it is often useful to consider multiple feature channels, for example, the color channels in images. For the general case of multiple input and output features, we extend the formulation of KA layers for $n$ elements with $d_{\text{in}}, d_{\text{out}}$ features by:

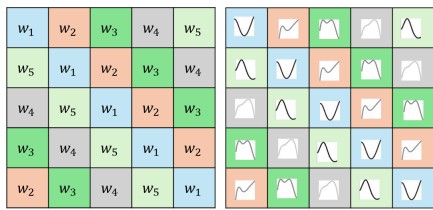

(a) Parameter-sharing (b) Function sharing

Figure 1: Parameter-sharing a and FS b for $C_5$ equivariant layers (1D-convolutional layers). Function sharing constrains the *functions* to be shared across the matrix according to the group action.

$$\Phi(\mathbf{x})_q = \sum_{p \in [n]} \Phi^{q,p}(\mathbf{x}_p), \quad q \in [n], \mathbf{x} \in \mathbb{R}^{n \times d_{\text{in}}}, \tag{4}$$

Where each $\Phi^{q,p} : \mathbb{R}^{d_{\text{in}}} \to \mathbb{R}^{d_{\text{out}}}$ is a KA layer. This creates a matrix of $n \times n$ *KA sub-layers*, similar to decomposing linear layers into sub-matrices. We use a similar formulation for the invariant case with $n_{\text{out}} = 1$. We show that, in this case, the functions are *externally* shared (figure 2a) across corresponding positions in the $\Phi^{q,p}$ sub-layers: (proof on Appx. B.6).

**Proposition 5.** *Let $\Phi$ be a G-equivariant (invariant) KA layer with $d_{in}, d_{out}$ input and output features. Then there exists KA layer $\hat{\Phi}$ s.t $\Phi, \hat{\Phi}$ represent the same function and the KA sub-layers satisfy : $\hat{\Phi}^{q,p} = \hat{\Phi}^{\sigma(q),\sigma(p)} \left( \hat{\Phi}^p = \hat{\Phi}^{\sigma(p)} \right), \ \forall \sigma \in G$. We call $\hat{\Phi}$ a G-equivariant (invariant) FS-KA layer.*

We note that this FS pattern follows the same sharing scheme as equivariant linear layers with multiple channels. These components allow us to define **G Function Sharing KA Networks ($G$-FS-KANs)** as finite compositions of $G$-equivariant and invariant FS-KA layers.

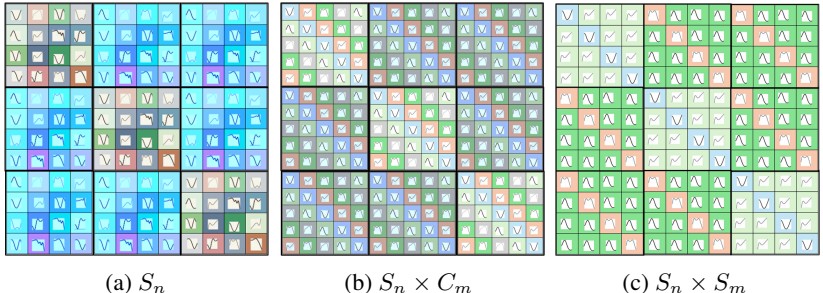

(a) $S_n$        (b) $S_n \times C_m$        (c) $S_n \times S_m$

Figure 2: Equivariant FS-KA layers for different groups. a is $S_3$-equivariant (with $d = 4$ feature channels), b is $S_3 \times C_5$-equivariant ($d = 1$) and c is $S_3 \times S_4$ equivariant ($d = 1$). The functions in each sub-layer in b are shared across the generalized diagonals (*internal* sharing w.r.t $C_4$), while the sub-layers themselves are shared according to equation 5 (*external* sharing w.r.t $S_3$).

**Efficient FS-KA layers.** In practice, standard KA layers can have significant computational and memory demands as they apply functions independently across all input and output pairs. However, in many cases, the sharing structure requires the same operations to be performed across multiple elements, followed by a sum aggregation. In linear layers, to reduce time and memory complexity, it can be helpful to commute sum or mean pooling with matrix multiplications. Inspired by this, we introduce the **Efficient FS-KA layer**, which first *aggregates* inputs according to the FS structure and *then* applies a shared KA sub-layer. While this relaxation does not yield equivalent layers, it reduces the number of nonlinear function applications while preserving equivariance, which can be easily verified. For example, consider a $S_n$-equivariant FS-KA layer (see equation 5). It applies a KA transformation independently to each element before performing sum-pooling over the set. In contrast, the efficient variant computes: $\tilde{\Phi}(\mathbf{x})_q = \tilde{\Phi}_1(\mathbf{x}_q) + \tilde{\Phi}_2\left(\sum_{p=1}^n \mathbf{x}_p\right)$. In this variant, we sum over *all* elements so the second term can be computed once and reused (*broadcast*) for other computations within the layer. While the efficient FS-KA layer has the same number of parameters, it reduces memory usage, especially during training, by retaining smaller computational graphs.

For arbitrary permutation groups $G$, efficient FS-KA layers are derived following the same principle: commuting element aggregations (sum or mean pooling) with application of shared functions according to the original sharing structure. In general, the efficiency of the layer is determined by the group structure. We discuss the efficiency and provide a more formal construction of efficient FS-KA layers for general permutation symmetry groups in Appx. A.3.

## 3.3 EXAMPLES AND IMPORTANT CASES

We now provide concrete examples of FS-KA layers and networks and demonstrate how they extend to important symmetry types. We first show how FS-KANs generalize previous equivariant KANs, and illustrate their enhanced interpretability through a synthetic learning example.

**FS-KANs generalize previous equivariant KANs.** For a set of $d$-dimensional vectors $\mathbf{x} \in \mathbb{R}^{n \times d}$, equivariant $S_n$-FS-KANs are composed of $S_n$-equivariant FS-KA layers in the form of:

$$\Phi(\mathbf{x})_q = \Phi_1(\mathbf{x}_q) + \sum_{p \neq q} \Phi_2(\mathbf{x}_p) \,, \tag{5}$$

where $\Phi_1, \Phi_2$ are shared KA sub-layers. We note that these layers are similarly structured to the DeepSets layers from Zaheer et al. (2017) and can be seen as a generalization of PointNet-KAN (Kashefi, 2024). We illustrate the layer structure on figure 2a. Similarly to the networks composed of equivalent linear layers, our $S_n$-FS-KAN can process sets with varying sizes, and the sum-pooling can be replaced with other invariant aggregations, such as mean or max, which preserve equivariance and might work better in practice. Another example is KA-convolution-based networks (Bodner et al., 2024), which are equivalent to our $C_n$-FS-KANs (Fig. 1b).

**Learning invariant formulas.** KANs are known for their superior interpretability through direct visualization of learned functions. To illustrate this in our case, we train both standard KAN and $S_n$-FS-KAN to learn $S_n$ invariant functions on synthetic data, following the methodology in Liu

et al. (2024). Figure 3 presents a visualization of the trained networks. Due to the function sharing scheme, $S_n$-FS-KAN learns shared spline functions across symmetric edges, making the equivariant structure immediately apparent and simplifying the learned network. In contrast, standard KAN learns independent spline functions for each edge, resulting in a more complex and less interpretable network that fails to respect underlying data symmetries. This demonstrates that FS-KAN not only maintains the interpretability benefits of standard KANs but enhances interpretability by making the equivariant structure explicit. We provide more examples in Appx. C.1.

We now discuss specific instantiations of FS-KA layers to two key symmetry types: direct-product symmetries (Maron et al., 2020; Wang et al., 2020) and high-order tensor symmetries (Maron et al., 2019b;c; Kondor et al., 2018; Keriven & Peyré, 2019).

**FS for direct-product symmetries.** For many data types that can be represented as matrices, the symmetry group can be formulated as a direct product of groups $G \times H$ where $G$ acts on the rows and $H$ acts on the columns. For example, in 2D images, symmetry exists in both the width dimension (horizontal shifts) and the height dimension (horizontal and vertical shifts). Similarly to Maron et al. (2020), $G \times H$ equivariant FS-KA layers have *external* sharing of KA *sublayers* (w.r.t $G$) and *internal* sharing of *functions*

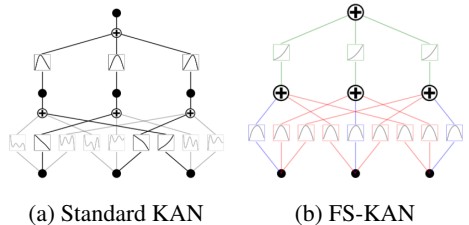

(a) Standard KAN      (b) FS-KAN

Figure 3: Visualization of learned spline functions for learning $f(x) = \exp(-x_1^2 - x_2^2 - x_3^2)$. FS-KAN b shares spline functions across symmetric edges (color-coded by function), with equivariant layers showing quadratic-like splines and the invariant output layer exhibiting exponential behavior, making the equivariant structure explicit and enhancing interpretability.

within each KA sub-layer (w.r.t to $H$). Figure 2b illustrates for $G = S_n, H = C_T$ how functions are shared *internally* within each sub-layer, forming a circulant structure and effectively acting as $1D$ KA-convolutions (Bodner et al., 2024), while the sub-layers themselves are shared *externally* within the overall layer. In figure 2c, we visualize the case when $G = S_n$ and $H = S_m$, as encountered in user-item interactions (Hartford et al., 2018). The formal construction and theoretical analysis are provided in Appx. A.1.

**FS for high-order tensors.** In many real-world applications, input data naturally appears as tensors of different orders $\mathbf{x} \in \mathbb{R}^{n^k}$, and the group action is the diagonal action, $(g \cdot \mathbf{x})_{i_1,\ldots,i_k} = \mathbf{x}_{g^{-1}(i_1),\ldots,g^{-1}(i_k)}$. For example, in graph-structured data, node features are first-order tensors, while connectivity can be represented as a second-order tensor. The KA layer formulation for layers mapping $k$-order tensors to a $k'$-order tensor can be readily derived from equation 4 and is given by: $\Phi(\mathbf{x})_{\mathbf{q}} = \sum_{\mathbf{p} \in [n]^k} \Phi^{\mathbf{q},\mathbf{p}}(\mathbf{x}_{\mathbf{p}}), \quad \mathbf{q} \in [n]^{k'}$, where each $\Phi^{\mathbf{q},\mathbf{p}} : \mathbb{R}^{d_{in}} \to \mathbb{R}^{d_{out}}$ are standard KA sublayers. In this case, the natural equivariant FS structure can be derived from how the group acts on tensor indices – $\Phi^{\mathbf{q},\mathbf{p}} = \Phi^{\sigma(\mathbf{q}),\sigma(\mathbf{p})}$ for all $\sigma \in G$, where $\sigma(\mathbf{i}) = (\sigma(i_1),\ldots,\sigma(i_k))$. This construction enables many important applications. For example, inspired by Maron et al. (2019b), we can build expressive GNNs Maron et al. (2019a) as well as networks for hypergraphs with applications including co-authorship networks (Feng et al., 2019; Yadati et al., 2019) and 3D mesh representation (Hajij et al., 2022). We provide more details and a discussion in Appx. A.2.

## 4 A THEORETICAL ANALYSIS OF THE EXPRESSIVE POWER OF FS-KANS

While equivariant linear layers are common building blocks, some of the resulting invariant and equivariant architectures, have known limitations – for example, cannot represent all invariant functions in the graph domain (Maron et al., 2019c; Chen et al., 2019; Geerts, 2020). In contrast, architectures like DeepSets (Zaheer et al., 2017) or PointNet (Qi et al., 2017) are universal for permutation-invariant functions for sets. In this section, we investigate the expressivity implications of choosing between FS-KANs and equivariant networks composed of equivariant linear layers. In particular, we show that for a specific permutation group $G$, both model families have the same expressive power in the uniform approximation sense. We begin by demonstrating that FS-KANs with splines activations can precisely implement any parameter-sharing MLP (within a bounded domain).

**Proposition 6.** *Let $\Omega$ be a bounded input domain, then for any parameter-sharing MLP $f$ with $l$ linear layers and ReLU activations, there exists FS-KAN $\Phi$ with at most $2l$ layers with splines activations such that:$f(\mathbf{x}) = \Phi(\mathbf{x}), \forall \mathbf{x} \in \Omega$.*

**Proof idea.** The key idea builds on Wang et al. (2024b), which shows that MLP layers with ReLU-like activations can be implemented using two KA layers: one for affine transformation and one for the point-wise ReLU. We decompose the parameter-sharing linear layer into sub-layers as in equation 4, each realizable by a KA layer. We use these realizations as the KA sub-layers, which compose an equivariant KA layer, and then apply Prop. 5 to obtain an equivalent FS-KA layer. Adding a point-wise FS-KA layer for the nonlinearity gives two FS-KA layers per MLP layer. Composing these yields an FS-KAN that reproduces the parameter-sharing MLP. The full proof is on Appx. B.7.

On the other hand, we will show next that parameter-sharing MLPs are capable of approximating FS-KANs to arbitrary precision (proof is on Appx. B.8) :

**Proposition 7.** *For any compact domain $\Omega$, $\epsilon > 0$, and FS-KAN $\Phi$ with $l$ layers and continuous activations, there exists a parameter-sharing MLP $f_\epsilon$ with at most $2l$ ReLU layers s.t. $||f - \Phi||_\infty < \epsilon$.*

Both proofs provide methods for building the corresponding networks, though not necessarily efficient in terms of depth or parameters. The results extend to higher-order tensors and direct products of symmetry groups via flattened tensor representations.

**Implications for the expressivity of FS-KANs.** We show that for a specific permutation group $G$, FS-KANs and parameter-sharing MLPs can approximate each other arbitrarily well, meaning both model families have equivalent expressive power in the uniform approximation sense. This equivalence has profound implications for both theoretical analysis and practical applications. It allows us to transfer known properties and guarantees between these model classes, creating a bridge between the established theory of parameter-sharing networks and KANs. For example, if DeepSets is proven universal for permutation-invariant functions, we can establish that the corresponding FS-KAN construction inherits this universality. As a direct implication of the above-mentioned expressive power equivalence, we establish the following corollary,

**Corollary 4.1.** *FS-KANs with splines activations inherit the approximation properties of parameter-sharing networks:*

1. *They are universal approximators for translation-equivariant functions (as established for CNNs by Yarotsky (2022)) and permutation-equivariant functions over sets (derived from Segol & Lipman (2019)).*

2. *Higher-order FS-KANs are universal approximators for $G$-invariant functions for any $G \leq S_n$ (Maron et al., 2019c).*

3. *FS-KANs for graphs involving $k$-order tensors have the expressive power of $k$-Invariant Graph Networks (Maron et al., 2019a; Geerts, 2020; Azizian & Lelarge, 2020) and hence the same discriminative power as the $k$-WL test (Morris et al., 2019).*

## 5 EXPERIMENTS

In this section, we evaluate the effectiveness of FS-KAN by comparing it to standard parameter-sharing MLPs across a range of tasks involving data with symmetries. Specifically, we aim to answer the following questions: (1) Do FS-KANs inherit the benefits of KANs? (2) How do FS-KANs perform compared to parameter-sharing MLPs and non-equivariant architectures? (3) Do FS-KANs offer advantages in low-data regimes? and (4) What is the trade-off between efficiency and expressivity when using the efficient FS-KAN variant (section 3.2)?

**Setup.** We evaluated FS-KANs on diverse invariant and equivariant tasks: signal classification with multiple measurements $(S_n \times C_T)$, point cloud classification $(S_n)$, and semi-supervised rating prediction $(S_n \times S_m)$. These tasks involve structured signals, 3D geometry, and user-item interactions. We compare FS-KANs and efficient FS-KANs against parameter-sharing MLP baselines, matching parameter counts when possible as done in recent

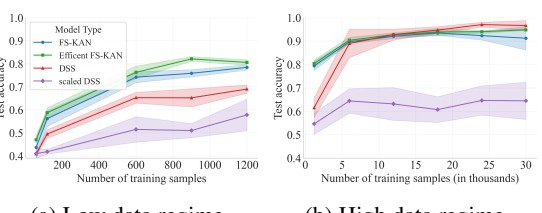

(a) Low data regime     (b) High data regime

Figure 4: Test accuracy for signal classification with multiple measurements with varying train size.

KAN literature (Liu et al., 2024; Bodner et al., 2024; Alter et al., 2024), and additionally include transformers and standard KANs for point cloud classification. While GCNNs (Cohen & Welling, 2016) could serve as baselines, they are computationally infeasible for these tasks. Performance is

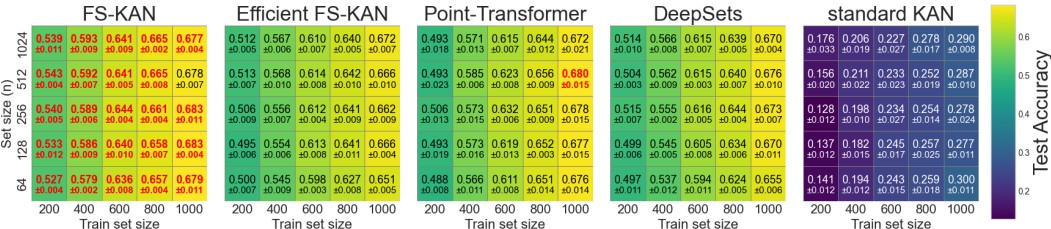

Figure 5: Test accuracy for point clouds classification with different numbers of $n$ points in each point cloud and varying train set size. The highest accuracy is shown in red and bold. Our FS-KAN consistently outperforms DeepSets in all configurations.

evaluated across varying training set sizes to examine behavior under different data regimes. All experiments use 5 random seeds with mean results and standard deviation error bars. Detailed task descriptions, datasets, and architectures are in Appx. C, with hyper-parameter study in Appx. D.

**Signal classification with multiple measurements** We follow the signal classification setup from Maron et al. (2020), using a synthetic dataset where each sample consists of a set of $n = 25$ noisy measurements of a periodic 1D signal sampled at $T = 100$ time steps. The clean signals belong to one of three classes—sine, sawtooth, or square wave—with varying amplitude, frequency, phase, and offset. The task is to classify the signal type given the noisy measurements. For this experiment, we use $S_n \times C_T$-FS-KAN and its efficient variant, composed of $S_n \times C_T$ equivariant and invariant FS-KA layers and batch-norms, with a total number of $3e4$ parameters. We compare our models to the parameter-sharing based *Deep Sets for Symmetric elements model* (DSS) proposed by the original work, which contains about $3e6$ parameters and scaled DSS with a comparable number of parameters to FS-KAN. Figure 4 shows the classification accuracy across different training set sizes. FS-KAN consistently outperforms both baseline models in the low-data regime (60-1200 examples) while yielding comparable results in higher-data regimes. The efficient variant achieves even higher accuracy than the full FS-KAN model while also reducing training time and memory usage. The training of the efficient variant was $\times 1.4$ faster than that of the full FS-KAN. However, the efficient variant was about 4 times slower than the DSS models (Table 3).

**Point cloud classification** We evaluate our models on the task of 3D object classification using the ModelNet40 dataset (Wu et al., 2015), which contains point cloud representations of objects from 40 categories such as *chair*, *table*, and *airplane*. Each sample consists of a set of $n$ points with $d = 3$ spatial coordinates. We evaluated the models' ability to generalize by varying the training set size and also examined how the number of points in each cloud affects their performance. We do not use data augmentations to ensure that the number of training examples reflects the actual dataset size. We compare $S_n$-FS-KAN and the efficient variant to parameter-sharing-based MLPs, composed of DeepSets (Zaheer et al., 2017) equivariant and invariant layers, as well as Point Transformer (Zhao et al., 2021) and a non-invariant KAN baseline. Each invariant model has approximately $5.5 \times 10^4$ parameters while the non-invariant KAN has varying parameter counts due to different input sizes. As observed in the previous experiment and supporting our hypothesis, FS-KAN outperforms the other models when the data is limited in both the number of samples and the number of points for each object, as depicted in figure 5. Consistent with previous works, the non-invariant KAN exhibits dramatically worse performance, highlighting the importance of designing symmetry-aware architectures. Additionally, we measure the runtime of each model in both training and inference (Table 4), as well as memory usage during both phases (Table 5). While the efficient variant is about $\times 1.5$ faster than the FS-KAN in training with larger $n$, it is still slower than DeepSets.

**Continual learning on point clouds** In real-world applications, data distributions frequently shift over time, necessitating models that can continuously adapt while preserving previously acquired knowledge (Wang et al., 2024a; Kirkpatrick et al., 2017). A critical challenge is *catastrophic forgetting*, where adapting to new distributions degrades performance on previous ones. To evaluate the robustness of our FS-KANs against catastrophic forgetting, we conducted a continual learning experiment following the methodology established in the original KAN paper and Park et al. (2024). Our experimental design comprises two phases: Phase 1 trains on the original ModelNet40 dataset (Wu et al., 2015) for point cloud classification (task A), while Phase 2 trains on a corrupted version featuring random translations and 3D rotations (task B). Further details appear in Appx. C.4. We

| Train Size | Model | acc. A1 ↑ | acc. A2 ↑ | acc. B2 ↑ | Forgetting ↓ | Avg Accuracy ↑ |
|---|---|---|---|---|---|---|
| 200 | FS-KAN | **0.531 ± 0.007** | **0.497 ± 0.014** | **0.343 ± 0.014** | **0.034 ± 0.018** | **0.420 ± 0.013** |
| 200 | Efficient FS-KAN | 0.511 ± 0.011 | 0.470 ± 0.018 | 0.320 ± 0.009 | 0.040 ± 0.010 | 0.395 ± 0.012 |
| 200 | DeepSets | 0.514 ± 0.009 | 0.455 ± 0.005 | 0.304 ± 0.014 | 0.059 ± 0.014 | 0.380 ± 0.009 |
| 400 | FS-KAN | **0.594 ± 0.008** | **0.550 ± 0.010** | **0.374 ± 0.014** | **0.044 ± 0.007** | **0.462 ± 0.012** |
| 400 | Efficient FS-KAN | 0.567 ± 0.004 | 0.506 ± 0.019 | 0.345 ± 0.015 | 0.061 ± 0.020 | 0.425 ± 0.013 |
| 400 | DeepSets | 0.562 ± 0.005 | 0.495 ± 0.019 | 0.344 ± 0.018 | 0.067 ± 0.017 | 0.419 ± 0.018 |
| 600 | FS-KAN | **0.635 ± 0.005** | **0.590 ± 0.014** | **0.411 ± 0.004** | 0.045 ± 0.013 | **0.501 ± 0.006** |
| 600 | Efficient FS-KAN | 0.613 ± 0.008 | 0.553 ± 0.014 | 0.384 ± 0.011 | 0.060 ± 0.012 | 0.469 ± 0.012 |
| 600 | DeepSets | 0.614 ± 0.007 | 0.559 ± 0.011 | 0.391 ± 0.006 | 0.055 ± 0.012 | 0.475 ± 0.007 |
| 800 | FS-KAN | **0.661 ± 0.006** | **0.623 ± 0.013** | **0.446 ± 0.012** | 0.038 ± 0.010 | **0.535 ± 0.011** |
| 800 | Efficient FS-KAN | 0.643 ± 0.007 | 0.586 ± 0.020 | 0.407 ± 0.015 | 0.056 ± 0.022 | 0.497 ± 0.016 |
| 800 | DeepSets | 0.641 ± 0.014 | 0.606 ± 0.005 | 0.427 ± 0.008 | **0.036 ± 0.011** | 0.516 ± 0.006 |
| 1000 | FS-KAN | **0.679 ± 0.007** | 0.640 ± 0.007 | 0.467 ± 0.011 | 0.040 ± 0.006 | 0.553 ± 0.009 |
| 1000 | Efficient FS-KAN | 0.666 ± 0.005 | 0.615 ± 0.003 | 0.435 ± 0.010 | 0.051 ± 0.006 | 0.525 ± 0.005 |
| 1000 | DeepSets | 0.670 ± 0.006 | **0.643 ± 0.016** | **0.468 ± 0.017** | **0.027 ± 0.020** | **0.555 ± 0.014** |

Table 1: Test accuracies and continual learning metrics on point cloud classification for different models and training set sizes. Best results are in bold.

evaluate performance using two metrics: *Forgetting*, calculated as the difference between accuracy on task A after Phase 1 and after Phase 2 (acc. A1 - acc. A2), and *Average Accuracy*, computed as the mean performance across both tasks after Phase 2. Lower forgetting indicates better retention of previously learned knowledge, while higher average accuracy reflects robust overall performance. The results in Table 1 show that FS-KAN demonstrates strong performance across training sizes, with clear advantages in low-data regimes where it consistently outperforms baselines.

**Semi-supervised rating prediction** We evaluate FS-KAN in a matrix completion task under extreme data scarcity. Given a partially observed user-item rating matrix, the goal is to predict the missing ratings. This setting exhibits a symmetry under both user and item permutations, which we model using $S_n \times S_m$ equivariant architectures. We run experiments on MovieLens-100K (Harper & Konstan, 2015) dataset and the sub-sampled Flixster, Douban and Yahoo Music presented by Monti et al. (2017). To simulate sparse supervision, we train on varying small fractions of the available training data. $S_n \times S_m$ FS-KAN and its efficient variant are compared to the Self-supervised Exchangeable Model (SSEM) proposed by Hartford et al. (2018) ($2e6$ parameters), composed of $S_n \times S_m$ equivariant parameter-sharing layers, and scaled SSEM with the same number of parameters as in FS-KAN ($9e4$ parameters). The data preparation procedure is detailed in Appx. C.5. As shown in figure 6, FS-KAN constantly achieves higher accuracy than the baselines in the low-data regime, highlighting its strong data efficiency. On larger datasets, the performance gap narrows, and the standard linear layer-based models become preferable due to their shorter training time.

**Discussion** Our experiments show that FS-KANs outperform parameter-sharing MLPs in low-data regimes while using fewer parameters, consistent with recent KAN findings (Liu et al., 2024; Kiamari et al., 2024). Additionally, our experiments show that our FS-KAN inherits the benefits of KAN, such as interpretability (figure 3, 7) and adaptability, while maintaining the benefit of incorporating symmetry. The efficient FS-KAN variant offers improved speed but remains slower than MLP baselines. These results suggest FS-KANs are most valuable for learning tasks on data with symmetries when data is limited, where their superior sample efficiency justifies the computational overhead.

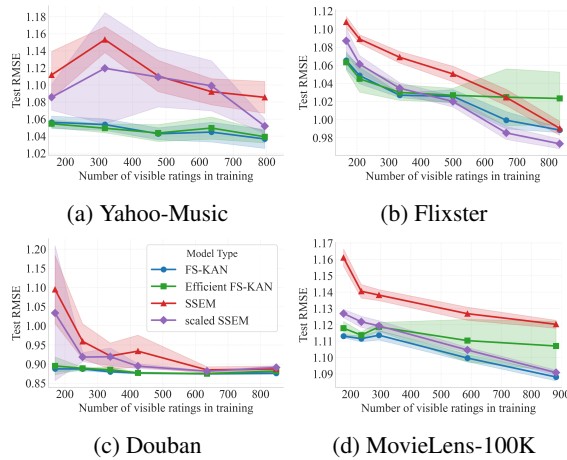

(a) Yahoo-Music  (b) Flixster

(c) Douban  (d) MovieLens-100K

Figure 6: Test RMSE comparison across different recommendation datasets. Our FS-KAN models outperform baseline approaches in the low data regime, demonstrating better data efficiency and generalization capability.

## 6 CONCLUSION

In this paper, we presented a novel approach to permutation-equivariant learning through KAN-based architectures. Our theoretical framework unifies several recent

architectures in this domain while providing new deep insights into their expressivity and offering a blueprint for designing invariant and equivariant KAN-based networks for numerous other symmetry groups currently lacking in the literature. While our proposed method demonstrates improved performance on tasks with limited data, it has several limitations. For example, the computational cost can be significant even with the efficient variant. This becomes especially apparent in the high-data regime. Hence, an important avenue for future work is to come up with faster implementations of FS-KANs. Finally, while our theoretical results focused mainly on expressivity, exploring properties of FS-KANs, such as generalization power, optimization issues, and scalability, is an important future research area.

## 7 ACKNOWLEDGMENTS

HM is supported by the Israel Science Foundation through a personal grant (ISF 264/23) and an equipment grant (ISF 532/23), and by the Career Advancement Chairs in Artificial Intelligence – Schmidt Futures. G.B. is supported by the Jacobs Qualcomm PhD Fellowship. FF conducted this work supported by an Aly Kaufman Post-Doctoral Fellowship.

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

## A  GENERALIZATIONS OF FS-KA LAYERS

### A.1  GENERALIZATION TO DIRECT PRODUCT SYMMETRIES

There are many cases where the symmetry of the data can be described using the direct product of groups $G \times H$. For example, in the case of a rating matrix $X \in \mathbb{R}^{n \times m}$ permutation $\sigma$ of the users and a permutation $\tau$ of the items will lead to corresponding permutations of rows and columns of the rating matrix. In general, for a $G \le S_n, H \le S_m$ the group $G \times H$ acts on $X \in \mathbb{R}^{n \times m \times d}$ via:

$$((\sigma, \tau) \cdot X)_{i,j} = X_{\sigma^{-1}(i), \tau^{-1}(j)}, \quad (\sigma, \tau) \in G \times H \tag{6}$$

The standard KA layer in this case can be formulated as:

$$\Phi(\mathbf{x})_q = \sum_{p=1}^{n} \Phi^{q,p}(\mathbf{x}_p) \tag{7}$$

Where each $\Phi^{q,p}$ is KA sub-layer from $\mathbb{R}^{m \times d_{\text{in}}}$ to $\mathbb{R}^{m \times d_{\text{out}}}$. As stated, $G \times H$ FS-KA layers can be constructed to be equivariant using $H$ FS-KA sub-layers as follows:

**Proposition 8.** *Let $\Phi : \mathbb{R}^{n \times m \times d_{in}} \to \mathbb{R}^{n \times m \times d_{out}}$ be $G \times H$ equivariant FS-KA layer. Then exists a KA layer $\hat{\Phi}$ composed of $H$-**equivariant FS-KA sub-layers** $\{\hat{\Phi}^{q,p}\}_{q,p \in [n]}$ with $d_{in}, d_{out}$ that satisfy:*

$$\hat{\Phi}^{q,p} = \hat{\Phi}^{\sigma(q),\sigma(p)}, \quad \forall \sigma \in G \tag{8}$$

*and,*

$$\hat{\Phi}_q(x) = \Phi(\mathbf{x})_q, \quad \forall \mathbf{x} \tag{9}$$

The proof is on Appx. B.9.

## A.2 HIGHER-ORDER TENSORS

As stated in section 3.3, the equivariant FS-KA layer for higher-order tensors satisfies:

$$\Phi^{\mathbf{q},\mathbf{p}} = \Phi^{\sigma(\mathbf{q}),\sigma(\mathbf{p})}, \quad \forall \sigma \in G. \tag{10}$$

FS-KA layers of this form can represent any other equivariant KA layer. The proof is similar to the first-order case, except it applies to $k$ and $k'$-tuples of indices. When the data involves mixed-order tensors, we can naturally use a superposition of FS-KA layers:

$$\Phi(\mathbf{x})_{k'} = \sum_k \Phi^{k',k}(\mathbf{x_k}), \quad \mathbf{q} \in [n]^{k'}, \tag{11}$$

where $\mathbf{x_k}$ is the k-order input tensor, $\Phi(\mathbf{x})_{k'}$ is a $k'$-order output tensor, and $\Phi^{k',k}$ is an FS-KA layer mapping from $k$ to $k'$-order tensors.

## A.3 EFFICIENT FS-KA LAYERS

In this section we will explain how to derive the efficient equivariant FS-KA layer for arbitrary group $G \leq S_n$. This derivation can be readily extended to invariant layers or to layers operating on higher-order tensors. In general, we can write the equivariant FS-KA layer output as follows (using Lemma B.2):

$$\Phi(\mathbf{x})_q = \sum_{p=1}^n \sum_{O_h \in [[n]^2/G]} \Phi^{(q,p)}(\mathbf{x}_p) \mathbb{I}\left\{(q,p) \in O_h\right\} = \sum_{p=1}^n \sum_{O_h \in [[n]^2/G]} \Phi^{(h)}(\mathbf{x}_p) \mathbb{I}\left\{(q,p) \in O_h\right\}, \tag{12}$$

where $[[n]^2/G]$ is the orbits of $[n]^2$ under the group action of $G$ and $\mathbb{I}$ is the indicator function, and $\Phi^{(h)}$ is a KA sub-layer shared across orbit $O_h$. Our *Efficient FS-KA* layer performs *sum-pooling* before applying *KA sub-layer*, i.e :

$$\tilde{\Phi}(\mathbf{x})_q = \sum_{O_h \in [[n]^2/G]} \Phi^{(h)}\left(\alpha_h \mathbf{x}_q + \sum_{p=1}^n \mathbf{x}_p \mathbb{I}\left\{(q,p) \in O_h\right\}\right), \tag{13}$$

where $\alpha_h \in \{0,1\}$ is for flexibility in the purpose of reusing computations (see the following examples).

While this relaxation does not have the expressivity guarantees of the FS-KA layer, it is still equivariant and can reduce the number of computations and memory costs.

**Equivariance.** We will show that the efficient variant is still $G$-equivariant. Let $\sigma \in G$, then:

$$\begin{aligned}
\tilde{\Phi}(\sigma \cdot \mathbf{x})_q &= \sum_{O_h \in [[n]^2/G]} \Phi^{(h)}\left(\alpha_h \mathbf{x}_{\sigma^{-1}(q)} + \sum_{p=1}^n \mathbf{x}_{\sigma^{-1}(p)} \mathbb{I}\left\{(q,p) \in O_h\right\}\right) \\
&= \sum_{O_h \in [[n]^2/G]} \Phi^{(h)}\left(\alpha_h \mathbf{x}_{\sigma^{-1}(q)} + \sum_{p=1}^n \mathbf{x}_p \mathbb{I}\left\{(q,\sigma(p)) \in O_h\right\}\right).
\end{aligned} \tag{14}$$

By the definition of the orbits, $(q, p) \in O_h$ iff $(\sigma(q), \sigma(p)) \in O_h$. Therefore,

$$
\begin{aligned}
\tilde{\Phi}(\sigma \cdot \mathbf{x})_q &= \sum_{O_h \in [[n]^2/G]} \Phi^{(h)}\left(\alpha_h \mathbf{x}_{\sigma^{-1}(q)} + \sum_{p=1}^n \mathbf{x}_p \mathbb{I}\left\{(\sigma^{-1}(q), p) \in O_h\right\}\right) \\
&= \tilde{\Phi}(\mathbf{x})_{\sigma^{-1}(q)} \\
&= (\sigma \cdot \tilde{\Phi}(\mathbf{x}))_q.
\end{aligned}
\tag{15}
$$

**Complexity analysis.** If the same orbit indicator is not zero for pairs $(q, p_1), (q, p_2)$, they must lie in the same orbit under the group action. Then, there must exist a permutation in $G$ that maps $p_1$ to $p_2$ while leaving $q$ unchanged. In other words, the relevant orbits of $[n]^2$ are determined by the action of the stabilizer subgroup $\text{Stab}_G(q)$, the set of elements in $G$ that fix $q$. Thus, for each $q$, the number of times $\Phi^{(h)}$ is applied in the efficient layer corresponds to the number of equivalence classes in $[n]/\text{Stab}_G(q)$.

As a result, the total number of function evaluations required by the Efficient FS-KA layer is at most:

$$
d_{\text{in}} d_{\text{out}} \sum_{q=1}^n |[n]/\text{Stab}_G(q)| \leq n^2 d_{\text{in}} d_{\text{out}}.
\tag{16}
$$

However, this is only an upper bound, as some computations can be reused. We will show how efficient this variant is in some examples.

**Examples.** Consider $G = S_n \times S_m$. In this case, we have four orbits, and the FS-KA layer can be expressed as

$$
\Phi(\mathbf{x})_{i,j} = \Phi_1(\mathbf{x}_{i,j}) + \sum_{k \neq j} \Phi_2(\mathbf{x}_{i,k}) + \sum_{l \neq i} \Phi_3(\mathbf{x}_{l,j}) + \sum_{l \neq i, k \neq j} \Phi_4(\mathbf{x}_{l,k}).
\tag{17}
$$

For $\alpha_2 = \alpha_3 = \alpha_4 = 1$ the efficient FS-KA layer has the form of:

$$
\tilde{\Phi}(\mathbf{x})_{i,j} = \tilde{\Phi}_1(\mathbf{x}_{i,j}) + \tilde{\Phi}_2\left(\sum_{k=1}^m \mathbf{x}_{i,k}\right) + \tilde{\Phi}_3\left(\sum_{l=1}^n \mathbf{x}_{l,j}\right) + \tilde{\Phi}_4\left(\sum_{l=1}^n \sum_{k=1}^m \mathbf{x}_{l,k}\right).
\tag{18}
$$

As the second and third term can be broadcast across the rows and columns respectively, and the fourth term is shared across all the layer, the layer performs only $(n + 3)d_{\text{in}} d_{\text{out}}$ complex computations.

We would also examine the case of $G = S_n \times C_T$. In this case, we have $2T$ orbits, and the layer can be expressed as

$$
\Phi(\mathbf{x})_{i,t} = \sum_{\tau=1}^T \Phi^{(1,\tau)}(\mathbf{x}_{i,t-\tau}) + \sum_{\tau=1}^T \sum_{k \neq i} \Phi^{(2,\tau)}(\mathbf{x}_{k,t-\tau}),
\tag{19}
$$

where $\Phi^{(1,\tau)}, \Phi^{(2,\tau)} : \mathbb{R}^{d_{\text{in}}} \to \mathbb{R}^{d_{\text{out}}}$ are shared KA sub-layers. We can reduce the number of distinct computations in the layer by a factor of $\frac{2n}{n+1}$ by,

$$
\tilde{\Phi}(\mathbf{x})_{i,t} = \sum_{\tau=1}^T \tilde{\Phi}^{(1,\tau)}(\mathbf{x}_{i,t-\tau}) + \sum_{\tau=1}^T \tilde{\Phi}^{(2,\tau)}\left(\sum_{k=i}^n \mathbf{x}_{k,t-\tau}\right).
\tag{20}
$$

In other words, the layer applies KA-convolution to each element and KA-convolution to the sum of all the elements in the set.

# B PROOFS

## B.1 KEY SUPPORTING LEMMAS

We state and prove two lemmas that will be used throughout the proofs of several of our claims.

**Lemma B.1.** *Let $f_1, \ldots, f_n : \mathbb{R}^{d_1} \to \mathbb{R}^{d_2}$ be a set of functions such that for any $x_1, \ldots, x_n \in \mathbb{R}^{d_1}$:*

$$\sum_{i=1}^{n} f_i(x_i) = 0. \tag{21}$$

*Then $f_1, \ldots f_n$ are constant functions.*

*Proof.* Without loss of generality, we will show for $j \in [n]$ that $f_j$ is constant. Let $x_1, \ldots, x_n \in \mathbb{R}^{d_1}$ and $y_1, \ldots, y_n \in \mathbb{R}^{d_1}$ such that:

$$x_i = y_i, \forall i \neq j. \tag{22}$$

By equation 21,

$$\sum_{i=1}^{n} f_i(x_i) - \sum_{i=1}^{n} f_i(y_i) = 0. \tag{23}$$

Using the condition on $x, y$ results in:

$$\begin{aligned}
0 = \sum_{i=1}^{n} f_i(x_i) - \sum_{i=1}^{n} f_i(y_i) &= \sum_{i \neq j} f_i(x_i) - \sum_{i \neq j} f_i(y_i) + f_j(x_j) - f_j(y_j) \\
&= \sum_{i \neq j} f_i(x_i) - \sum_{i \neq j} f_i(x_i) + f_j(x_j) - f_j(y_j) \\
&= f_j(x_j) - f_j(y_j).
\end{aligned} \tag{24}$$

Therefore:

$$f_j(x_j) = f_j(y_j), \quad \forall x_j, y_j \in \mathbb{R}. \tag{25}$$

Meaning $f_j$ is constant. $\qquad\square$

**Lemma B.2.** *Let $G \leq S_n$ acting on $[n]^2$ and $O_1, \ldots, O_H$ be the orbits of $[n]^2$ under the group action of $G$. Then for any $A \in \mathbb{R}^{n \times n \times d}$:*

$$\sum_{j=1}^{n} A_{i,j} = \sum_{j=1}^{n} \sum_{h=1}^{H} A_{i,j} \mathbb{I}\{(i,j) \in O_h\}, \tag{26}$$

*where $\mathbb{I}$ is the indicator function.*

*Proof.* Each $(i, j)$ belongs to exactly one orbit, therefore:

$$\sum_{h=1}^{H} A_{i,j} \mathbb{I}\{(i,j) \in O_h\} = A_{i,j}, \tag{27}$$

which concludes the proof. $\qquad\square$

## B.2 PROOF FOR PROP. 1

*Proof.* We need to prove $\Phi$ is equivariant, i.e:

$$\sigma \circ \Phi = \Phi \circ \sigma, \ \forall \sigma \in G. \tag{28}$$

For convenience, we will show the equivalent condition:

$$\Phi = \sigma^{-1} \circ \Phi \circ \sigma, \ \forall \sigma \in G. \tag{29}$$

Let $\mathbf{x} \in \mathbb{R}^n$ be an input vector. By definition, the output of the layer is given by:

$$\Phi(\mathbf{x})_q = \sum_{p=1}^{n} \phi_{q,p}(x_p), \forall q \in [n]. \tag{30}$$

Applying permutation $\sigma \in G$ in the input space:

$$\Phi(\sigma \cdot \mathbf{x})_q = \sum_{p=1}^{n} \phi_{q,p}(x_{\sigma^{-1}(p)}) = \sum_{p=1}^{n} \phi_{q,\sigma(p)}(x_p), \tag{31}$$

where the last transition is valid since $\sigma$ is a bijection, and we change the order of the summation. Applying the inverse permutation on the output results in

$$(\sigma^{-1}\Phi(\sigma \cdot \mathbf{x}))_q = (\Phi(\sigma \cdot \mathbf{x}))_{\sigma(q)} = \sum_{p=1}^{n} \phi_{\sigma(q),\sigma(p)}(x_p). \tag{32}$$

The last transition uses equation 31. By the function sharing condition equation 2,

$$(\sigma^{-1}\Phi(\sigma x))_q = \sum_{p=1}^{n} \phi_{\sigma(q),\sigma(p)}(x_p) = \sum_{p=1}^{n} \phi_{q,p}(x_p) = \Phi(x)_q. \tag{33}$$

Therefore, $\Phi$ is equivariant. $\qquad\square$

### B.3 Proof for Prop. 2

*Proof.* To make the proof more concrete, we illustrate it using the case $G = S_n$, and encourage the reader to refer to Example 3 to see how the argument aligns with that specific case.
Let $O_1, O_2, \ldots, O_H$ denote the orbits of $[n]^2$ under the action of $G$, where $H = |[n]^2/G|$ is the number of orbits. Let $(q_h, p_h)$ be a representative of orbit $O_h$ for each $h \in [H]$.
In the example, we have $H = 2$. The first orbit consists of the diagonal entries $O_1 = \{(i,i) \mid i \in [n]\}$, with $(q_1, p_1) = (1,1)$ as the representative. The second orbit includes all off-diagonal entries $O_2 = \{(i,j) \mid i, j \in [n],\ i \neq j\}$, with $(q_2, p_2) = (1,2)$ as a representative.
Since $\Phi$ is $G$-equivariant, we have:

$$\sigma^{-1}\Phi(\sigma \cdot \mathbf{x}) - \Phi(\mathbf{x}) = 0, \quad \forall \sigma \in G. \tag{34}$$

Using Eq. equation 32, this condition simplifies to:

$$\begin{aligned}
0 &= (\sigma^{-1}\Phi(\sigma \cdot \mathbf{x}))_q - \Phi(\mathbf{x})_q \\
&= \sum_{p=1}^{n} \phi_{\sigma(q),\sigma(p)}(x_p) - \sum_{p=1}^{n} \phi_{q,p}(x_p) \\
&= \sum_{p=1}^{n} (\phi_{\sigma(q),\sigma(p)}(x_p) - \phi_{q,p}(x_p)), \quad \forall \mathbf{x}.
\end{aligned} \tag{35}$$

Using Lemma B.1, each term in the sum must be constant. Thus, functions with indices from the same orbit differ only by a constant. For any orbit $O_h$ and for any $(q, p) \in O_h$, we define:

$$C_{q,p} = \phi_{q,p}(\cdot) - \phi_{q_h,p_h}(\cdot). \tag{36}$$

In the $S_n$ case, it means that:

$$\begin{aligned}
C_{2,2} = \phi_{2,2}(x) - \phi_{1,1}(x) \ ; \ C_{3,3} = \phi_{3,3}(x) - \phi_{1,1}(x) \ ; \ldots \\
C_{2,3} = \phi_{2,3}(x) - \phi_{1,2}(x) \ ; \ C_{1,3} = \phi_{1,3}(x) - \phi_{1,2}(x) \ ; \ldots
\end{aligned} \tag{37}$$

We also define $\alpha_1, \ldots, \alpha_q$ as:

$$\alpha_q \triangleq \sum_{p=1}^{n} C_{q,p}, \ \ q \in [n]. \tag{38}$$

Substitute equation 36 in equation 35 results in:

$$
\begin{aligned}
0 &= \sum_{p=1}^{n} \left( \phi_{\sigma(q),\sigma(p)}(x_p) - \phi_{q,p}(x_p) \right) \\
&= \sum_{p=1}^{n} \left( \phi_{\sigma(q),\sigma(p)}(x_p) - \phi_{q_h,p_h}(x_p) + \phi_{q_h,p_h}(x_p) - \phi_{q,p}(x_p) \right) \\
&= \sum_{p=1}^{n} C_{\sigma(q),\sigma(p)} - C_{q,p} \\
&= \sum_{p=1}^{n} C_{\sigma(q),\sigma(p)} - \sum_{p=1}^{n} C_{q,p} \\
&= \sum_{p=1}^{n} C_{\sigma(q),p} - \sum_{p=1}^{n} C_{q,p} \\
&= \alpha_{\sigma(q)} - \alpha_q.
\end{aligned}
\tag{39}
$$

Where in the second last transition, we used the fact $\sigma$ is a bijection, and we can change the order of the sum.

equation 39 suggests that:

$$
\alpha_q = \alpha_{\sigma(q)}, \forall \sigma \in G.
\tag{40}
$$

In the example it means $\alpha_1 = \alpha_2 = \dots$ as all the set $[n]$ is in the same orbit.

Now, define $\hat{\Phi}$ to be a KA layer such that for any $(q,p) \in O_h$:

$$
\hat{\phi}_{q,p}(\cdot) \triangleq \phi_{q_h,p_h}(\cdot) + \frac{1}{n}\alpha_{q_h}
\tag{41}
$$

Since this definition depends only on orbit representative, $\hat{\Phi}$ is a $G$-equivariant FS-KA layer, i.e $\hat{\phi}_{q,p} = \hat{\phi}_{\sigma(q),\sigma(p)}$ for all $\sigma \in G$. To show that $\Phi$ and $\hat{\Phi}$ represent the same function we will use Lemma B.2:

$$
\begin{aligned}
\Phi(\mathbf{x})_q &= \sum_{p=1}^{n} \phi_{q,p}(x_p) \\
&= \sum_{p=1}^{n} \sum_{h=1}^{H} \phi_{q,p}(x_p) \mathbb{I}\{(q,p) \in O_h\} \\
&= \sum_{p=1}^{n} \sum_{h=1}^{H} (\phi_{q_h,p_h}(x_p) + C_{q,p}) \mathbb{I}\{(q,p) \in O_h\},
\end{aligned}
\tag{42}
$$

where $\mathbb{I}$ is the indicator function. Substitute equation 41 into the equation,

$$
\begin{aligned}
\cdots &= \sum_{p=1}^{n} \sum_{h=1}^{H} (\hat{\phi}_{q,p}(x_p) - \frac{1}{n}\alpha_{q_h} + C_{q,p}) \mathbb{I}\{(q,p) \in O_h\} \\
&= \sum_{p=1}^{n} \sum_{h=1}^{H} \hat{\phi}_{q,p}(x_p) \mathbb{I}\{(q,p) \in O_h\} - \frac{1}{n} \sum_{p=1}^{n} \sum_{h=1}^{H} \alpha_{q_h} \mathbb{I}\{(q,p) \in O_h\} + \sum_{p=1}^{n} \sum_{h=1}^{H} C_{q,p} \mathbb{I}\{(q,p) \in O_h\}.
\end{aligned}
\tag{43}
$$

We can simplify the first term using Lemma B.2:

$$
\sum_{p=1}^{n} \sum_{h=1}^{H} \hat{\phi}_{q,p}(x_p) \mathbb{I}\{(q,p) \in O_h\} = \sum_{p=1}^{n} \hat{\phi}_{q,p}(x_p) = \hat{\Phi}(\mathbf{x})_q.
\tag{44}
$$

Furthermore, we can use the lemma to simplify the third term:

$$
\sum_{p=1}^{n} \sum_{h=1}^{H} C_{q,p} \mathbb{I}\{(q,p) \in O_h\} = \sum_{p=1}^{n} C_{q,p} = \alpha_q.
\tag{45}
$$

Note that if $(q, p) \in O_h$, there must exist a permutation that maps $q$ to $q_h$. Therefore by equation 40 it implies that $\alpha_q = \alpha_{q_h}$. Then the second term can also be simplified as:

$$\frac{1}{n} \sum_{p=1}^{n} \sum_{h=1}^{H} \alpha_{q_h} \mathbb{I}\{(q, p) \in O_h\} = \frac{1}{n} \sum_{p=1}^{n} \sum_{h=1}^{H} \alpha_q \mathbb{I}\{(q, p) \in O_h\} = \frac{1}{n} \sum_{p=1}^{n} \alpha_q = \alpha_q. \qquad (46)$$

Combining it all, we get:

$$\Phi(\mathbf{x})_q = \hat{\Phi}(\mathbf{x})_q - \alpha_q + \alpha_q = \hat{\Phi}(\mathbf{x})_q. \qquad (47)$$

Therefore, $\Phi$ and $\hat{\Phi}$ compute the same function, completing the proof. $\qquad \square$

### B.4 PROOF FOR PROP. 3

*Proof.* The output of the layer is given by:

$$\Phi(\mathbf{x}) = \sum_{p=1}^{n} \phi_p(x_p) \qquad (48)$$

Applying permutation $\sigma \in G$ in the input space:

$$\Phi(\sigma \cdot \mathbf{x}) = \sum_{p=1}^{n} \phi_p(x_{\sigma^{-1}(p)}) = \sum_{p=1}^{n} \phi_{\sigma(p)}(x_p) \qquad (49)$$

By the invariant FS condition ($\phi_{\sigma(p)} = \phi_p$)

$$\Phi(\sigma \cdot \mathbf{x}) = \sum_{p=1}^{n} \phi_{\sigma(p)}(x_p) = \sum_{p=1}^{n} \phi_p(x_p) = \Phi(\mathbf{x}) \qquad (50)$$

Which concludes the proof. $\qquad \square$

### B.5 PROOF FOR PROP. 4

*Proof.* Denote $\Phi^{(bc)} : \mathbb{R}^n \to \mathbb{R}^n$ as a KA layer such that:

$$\Phi^{(bc)}_{q,p} = \Phi_p \qquad (51)$$

i,e. $\Phi$ is broadcast $n$ times.
$\Phi$ is $G$-invariant, then by the definition of $\Phi^{(bc)}$:

$$\Phi^{(bc)}(\sigma \cdot \mathbf{x})_q = \Phi(\sigma \cdot \mathbf{x}) = \Phi(\mathbf{x}), \quad \forall \sigma \in G \qquad (52)$$

On the other hand,

$$(\sigma \cdot \Phi^{(bc)}(\mathbf{x}))_q = \Phi^{(bc)}(\mathbf{x})_{\sigma^{-1}(q)} = \Phi(\mathbf{x}) \qquad (53)$$

Therefore, $\Phi^{(bc)}$ is an $G$-equivariant layer. by Proposition 2 there exist $G$-equivariant FS-KA layer $\hat{\Phi}^{(bc)}$ that is equivalent to $\Phi^{(bc)}$. Now, define $\hat{\Phi}$ as:

$$\hat{\Phi}(x) = \hat{\Phi}^{(bc)}(x)_1 \qquad (54)$$

By the definition of $\Phi^{(bc)}$, $\hat{\Phi}^{(bc)}(x)_1 = \Phi(x)$. Therefore, the layers are equivalent.
Using the equivariant FS condition for $\hat{\Phi}^{(bc)}$ we can show that for any $p \in [n]$ and $\sigma \in G$:

$$\hat{\phi}_{\sigma(p)} = \hat{\phi}^{(bc)}_{1,\sigma(p)} = \hat{\phi}^{(bc)}_{\sigma^{-1}(1),\sigma^{-1}(\sigma(p))} = \hat{\phi}^{(bc)}_{\sigma^{-1}(1),p} = \hat{\phi}^{(bc)}_{1,p} = \hat{\phi}_p \qquad (55)$$

Therefore, $\hat{\Phi}$ is $G$-invariant FS-KA layer. $\qquad \square$

### B.6 PROOF FOR PROP. 5

The proof of this proposition follows exactly the same structure as the proofs of Prop. 2, 4 only with $x_i$ as a vector of dimension $d_{\text{in}}$ and $\phi_{q,p}$ replaced with $\Phi^{q,p}$.

### B.7 PROOF FOR PROP. 6

*Proof.* We begin by recalling Theorem 3.2 from Wang et al. (2024b) establishes that each layer of an MLP with activation function $\sigma_k(\cdot) = max(0, \cdot)^k$ can be represented by a KAN with two hidden layers and grid size $G = 2$ with degree $k$ B-splines. Here, $k$ determines the polynomial degree of both the activation and the spline basis functions, while $G$ specifies the number of grid intervals used in the spline parameterization. Their proof demonstrates this by representing the linear part of an MLP layer using a single KAN layer and the non-linear activation using another KAN layer with a diagonal structure. i,e for a bounded domain $\Omega \subset \mathbb{R}^{d_{in}}$ and an affine layer $L : \mathbb{R}^{d_{in}} \to \mathbb{R}^{d_{out}}$ exists KA layers $\Phi : \mathbb{R}^{d_{in}} \to \mathbb{R}^{d_{out}}$ and $\Psi : \mathbb{R}^{d_{out}} \to \mathbb{R}^{d_{out}}$ such that:

$$\begin{aligned} \Phi(\mathbf{x}) &= L(\mathbf{x}), \ \ \forall \mathbf{x} \in \Omega \\ \Psi(\mathbf{x}) &= \sigma_k(\mathbf{x}), \ \ \forall \mathbf{x} \in \Phi(\Omega) \end{aligned} \tag{56}$$

In the general case, the MLP has equivariant layers and invariant layer followed by a standard MLP. As the work of Wang et al. (2024b) already handles standard MLPs, we will focus on showing that any equivariant or invariant linear layer with $\sigma_k$ activation can be represented using one hidden layer FS-KAN.

Let $L : \Omega \to \mathbb{R}^{n \times d_{out}}$ be an equivariant affine layer. By Wang et al. (2024b) work exists a KA layer such that:

$$\Phi(\mathbf{x}) = L(\mathbf{x}), \forall \mathbf{x} \in \Omega \tag{57}$$

$\Phi$ is equivariant but not necessary FS layer. By Proposition 5 exists FS-KA layer $\hat{\Phi}$ such that:

$$\Phi(\mathbf{x}) = \hat{\Phi}(\mathbf{x}) \tag{58}$$

We will use $\Psi$ ( defined on equation 56 ) to construct KA layer $\bar{\Psi}$ that simulate the ReLU activation:

$$\bar{\Psi}^{q,p} = \begin{cases} \Psi, & q = p \\ 0, & q \neq p \end{cases} \tag{59}$$

Note that $\bar{\Psi}$ acts element-wise, thus $S_n$-equivariant and therefore $G$-equivariant as well. Furthermore, $\bar{\Psi}$ is a FS-KA layer.

Thus, $\bar{\Psi} \circ \hat{\Phi}$ is an FS-KAN with one hidden layer that simulates $\sigma_k \circ L$ on a bounded domain. The full network is realizable by simply composing the realizations of each layer while considering the previous layer's image to be the current layer's input domain. $\square$

### B.8 PROOF FOR PROP. 7

*Proof.* We will prove the claim for a single equivariant layer $\Phi$. The invariant layer proof follows the same proof structure. By definition, a parameter-sharing equivariant layer can be written as:

$$\Phi(\mathbf{x})_q = \sum_{p=1}^{n} \Phi^{(q,p)}(\mathbf{x}_p) \tag{60}$$

where each $\Phi^{(q,p)}$ is a function $\mathbb{R}^{d_{in}} \to \mathbb{R}^{d_{out}}$, and the functions satisfy the equivariance FS condition:

$$\Phi^{(q,p)} = \Phi^{(\sigma(q),\sigma(p))}, \quad \forall \sigma \in G \tag{61}$$

Let $\epsilon > 0$. By the universal approximation theorem Hornik et al. (1989) for MLPs with ReLU activations, for each function $\Phi^{(q,p)}$ (which is continuous on a compact domain), there exists a two-layer MLP $f^{(q,p)}$ with ReLU activations such that:

$$\left\| \Phi^{(q,p)} - f^{(q,p)} \right\|_{\infty} < \frac{\epsilon}{n^2}$$

We assume all $f^{(q,p)}$ have the same width $d_{max}$ by padding with zeros if needed. We denote $W_1^{(q,p)}, b_1^{(q,p)}$ as the weights and biases of the first input layer and $W_2^{(q,p)}, b_2^{(q,p)}$ as those of the output linear layer.

Because $\Phi^{(q,p)} = \Phi^{(\sigma(q),\sigma(p))}$, we may assume that $f^{(q,p)} = f^{(\sigma(q),\sigma(p))}$ and therefore their parameters are shared accordingly:

$$W_1^{(q,p)} = W_1^{(\sigma(q),\sigma(p))}, \quad W_2^{(q,p)} = W_2^{(\sigma(q),\sigma(p))}, \text{ etc.} \tag{62}$$

We now construct a two-layer MLP approximating $\Phi(x)$. the first layer $L_1 : \mathbb{R}^{n \times d_{in}} \to \mathbb{R}^{n^2 \times d_{max}}$ construction defined by:

$$L_1(\mathbf{x})_{q,p} = W_1^{(q,p)}\mathbf{x}^{(p)} + b_1^{(q,p)} \tag{63}$$

We will show this is indeed an equivariant linear map:

$$L_1(\sigma \cdot \mathbf{x})_{q,p} = W_1^{(q,p)}\mathbf{x}^{(\sigma^{-1}(p))} + b_1^{(q,p)} \tag{64}$$

On the other hand,

$$(\sigma \cdot L_1(\mathbf{x}))_{q,p} = (L_1(\mathbf{x}))_{\sigma^{-1}(q),\sigma^{-1}(p)} = W_1^{\sigma^{-1}(q),\sigma^{-1}(p)}\mathbf{x}^{(\sigma^{-1}(p))} + b_1^{\sigma^{-1}(q),\sigma^{-1}(p)} \tag{65}$$

Using equation 62:

$$(\sigma \cdot L_1(\mathbf{x}))_{q,p} = W_1^{(q,p)}\mathbf{x}^{(\sigma^{-1}(p))} + b_1^{(q,p)} = L_1(\sigma \cdot \mathbf{x})_{q,p} \tag{66}$$

$L_1$ is an equivariant linear map, so it must be a parameter-sharing linear layer (Wood & Shawe-Taylor, 1996). Second layer construction $L_2 : \mathbb{R}^{n^2 \times d_{max}} \to \mathbb{R}^{n \times d_{out}}$ will be defined as:

$$L_2(x)_q = \sum_{p=1}^{n} W_2^{(q,p)}x_{q,p} + b_2^{(q,p)} \tag{67}$$

Similarly, $L_2$ is also equivariant linear map, therefore has a parameter sharing structure. The parameter sharing MLP $f$ is given by:

$$f(x) = L_2[L_1(x)]_+ \tag{68}$$

Where $[\cdot]_+$ is the ReLU activation. By the construction of $f$:

$$f(\mathbf{x})_q = \sum_{p=1}^{n} W_2^{(q,p)}[W_1^{(q,p)}\mathbf{x}^{(p)} + b_1^{(q,p)}]_+ + b_2^{(q,p)} = \sum_{p=1}^{n} f^{(q,p)}(\mathbf{x}_p) \tag{69}$$

In total, the approximation error of the layer would be $\epsilon$ as:

$$\begin{aligned}
||\Phi(x) - f(x)|| &= \sum_q ||\Phi(x)_q - f(x)_q|| \\
&= \sum_q ||\sum_p \Phi^{(q,p)}(x_p) - f^{(q,p)}(x_p)|| \\
&\leq \sum_q \sum_p ||\Phi^{(q,p)}(x_p) - f^{(q,p)}(x_p)|| \\
&= n^2 \sum_q \sum_p \epsilon/n^2 \\
&= \epsilon
\end{aligned} \tag{70}$$

Using Lemma 6 from Lim et al. (2022) (*Layer-wise universality implies universality*), using composition of the MLPs approximation for each layer approximates the whole MLP up to any precision. $\square$

### B.9 PROOF FOR PROP. 8

*Proof.* Let $P : [n] \times [m] \to [nm]$ be a bijection. We define the flatten form of $\mathbf{x} \in \mathbb{R}^{n \times m \times d}$ as:

$$Vec(\mathbf{x})_i = \mathbf{x}_{P^{-1}(i)} \tag{71}$$

The group action of $G \times H$ extend naturally as:

$$\begin{aligned}
(g,h) \cdot P(i,j) &= P((g,h) \cdot (i,j)) = P(g(i),h(j)), \quad (g,h) \in G \times H \\
((g,h) \cdot Vec(\mathbf{x})) &= Vec((g,h) \cdot \mathbf{x})
\end{aligned} \tag{72}$$

We will define the KA layer $\bar{\bar{\Phi}}$ on the flattened vector in an implicit way:

$$\Phi^{q_1,p_1,q_2,p_2} = \bar{\bar{\Phi}}^{P_1(q_1,q_2),P(p_1,p_2)} \tag{73}$$

On the one hand,

$$
\begin{aligned}
\Phi(x)_{q_1,q_2} &= \sum_{p_1=1}^{n} \sum_{p_2=1}^{n} \Phi^{q_1,p_1,q_2,p_2}(x_{p_1,p_2}) \\
&= \sum_{p_1=1}^{n} \sum_{p_2=1}^{n} \bar{\bar{\Phi}}^{P_1(q_1,q_2),P(p_1,p_2)}(x_{p_1,p_2}) \\
&= \sum_{p_1=1}^{n} \sum_{p_2=1}^{n} \bar{\bar{\Phi}}^{P_1(q_1,q_2),P(p_1,p_2)}(Vec(x)_{P(p_1,p_2)}) \\
&= \bar{\bar{\Phi}}(Vec(x))_{P_1(q_1,q_2)}
\end{aligned}
\tag{74}
$$

On the other hand,

$$\Phi(x)_{q_1,q_2} = Vec(\Phi(x))_{P(q_1,q_2)} \tag{75}$$

Therefore,

$$\bar{\bar{\Phi}}(Vec(x)) = Vec(\Phi(x)) \tag{76}$$

Specifically, $\bar{\bar{\Phi}}$ is $G \times H$ equivariant, then by Prop. 5 exists an equivalent equivariant FS-KA layer $\tilde{\Phi}$. Note that $G \times H$ acts only on the first coordinate, and therefore, the use of our proposition is justified. By the FS condition:

$$\tilde{\Phi}^{i,j} = \tilde{\Phi}^{\sigma(i),\sigma(j)}, \forall \sigma \in G \times H \tag{77}$$

We will construct the equivalent KA layer $\bar{\bar{\Phi}}$ for the tensor form by:

$$\hat{\Phi}^{i_1,i_2,j_1,j_2} = \tilde{\Phi}^{P(i_1,j_1),P(i_2,j_2)} \tag{78}$$

This construction implies too that $\tilde{\Phi}(Vec(x)) = Vec(\hat{\Phi}(x))$ By the FS condition and by construction of $\hat{\Phi}$ we get:

$$
\begin{aligned}
\hat{\Phi}^{g(i_1),g(i_2),h(j_1),h(j_2)} &= \tilde{\Phi}^{P(g(i_1),h(j_1)),P(g(i_2),h(j_2))} \\
&= \tilde{\Phi}^{P(i_1,j_1),P(i_2,j_2)} \\
&= \bar{\bar{\Phi}}^{i_1,i_2,j_1,j_2}, \quad \forall(g,h) \in G \times H
\end{aligned}
\tag{79}
$$

For $h = e_H$ (identity of $H$), we prove the sharing of the KA-sub layers, and for $g = e_G$, we prove the sharing within each sub-layer. $\qquad\square$

## C    Implementation details

All experiments on were implemented using the PyTorch framework and trained using the AdamW optimizer (Loshchilov & Hutter, 2017). The FS-KAN variants implementations are based on the `Efficient-KAN` (Blealtan, 2024) package and the ConvKAN (Bodner et al., 2024) GitHub repository, with the exception of the symbolic formula experiments, which used the PyKAN library (Liu et al., 2024) and LBFGS optimizer for both KAN and FS-KAN implementations. For MLP-based models, we applied $\ell_2$ weight regularization; for KAN-based models, we used the regularization provided by the `Efficient-KAN` library. The exact parameter count of the invariant/equivariant models are in Table 2. As for the non-invariant standard KAN in the point cloud classification task, the parameter counts vary between 42,240 and 503,040 depending on the specific configuration. The regularization term was scaled by a hyper-parameter $\eta$ in both cases. Models were trained on NVIDIA V100 GPUs. We trained the models with 5 different seeds in all experiments and configurations.

While Group Convolutional Neural Networks (GCNNs) Cohen & Welling (2016) can also be considered as a baseline, they become computationally intractable for the large symmetry groups we consider, as group convolutions scale with the size of the group. This illustrates how our approach is more practical for real-world applications.

| Task | FS-KAN & Efficient FS-KAN | Scaled Baseline | Standard Baseline |
|------|---------------------------|-----------------|-------------------|
| Signal classification | 33,834 | 35,467 (SSEM) | 3,234,643 (SSEM) |
| Point cloud classification | 55,584 | 55,976 (DeepSets) | 55,159 (Point Transformer) |
| Recommendation system | 88,608 | 90,017 (DeepSets) | 2,114,309 (DeepSets) |

Table 2: Parameter count comparison across different tasks and methods

## C.1 LEARNING INVARIANT SYMBOLIC FORMULAS

The dataset generation follows the procedure provided in Liu et al. (2024). FS-KAN models consist of one equivariant layer with single feature channels followed by one invariant layer. While the FS-KAN is illustrated for $n = 3$, it can handle varying input sizes. For the baseline KAN, we use one hidden layer of width 3. Both models use L-BFGS optimization with 25 steps and $\lambda = 0.001$. We refer to figure 7 for more results on different invariant formulas.

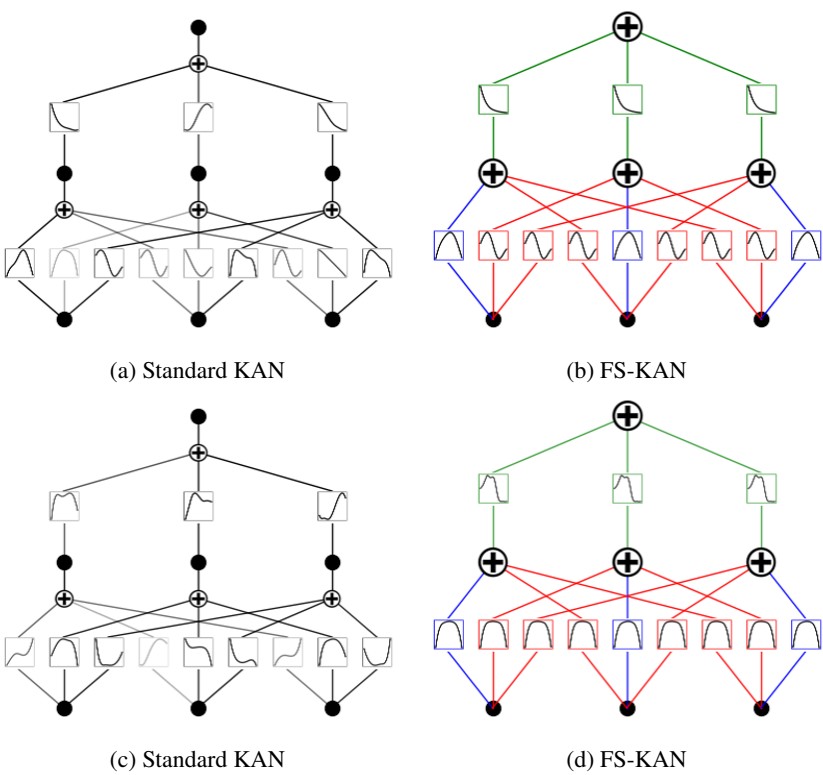

(a) Standard KAN        (b) FS-KAN

(c) Standard KAN        (d) FS-KAN

Figure 7: Visualization of learned spline functions. Top row: $f(x) = e^{\frac{10x_1^2 + \sin(\pi x_2) + \sin(\pi x_3)}{3}} + e^{\frac{10x_2^2 + \sin(\pi x_1) + \sin(\pi x_3)}{3}} + e^{\frac{10x_3^2 + \sin(\pi x_1) + \sin(\pi x_2)}{3}}$. Bottom row: $f(x) = \tanh(5(x_1^4 + x_2^4 + x_3^4) - 1)$.

## C.2 SIGNAL CLASSIFICATION EXPERIMENT

**Data preparation.** The dataset generation follows the procedure described on Maron et al. (2020), where each clean signal was randomly selected from a predefined set of types - sine, square, and sawtooth with $T = 100$ time steps. To construct a set of noisy measurements, each signal is duplicated multiple times, adding independent Gaussian noise to each copy. All samples were generated using the code provided by Maron et al. (2020). Validation and test sets have a fixed size of 300 samples each.

**Networks and training.** FS-KAN and efficient FS-KAN models have 3 hidden layers of width $16, 16, 8$, while the baseline DSS used hidden layers of width $160, 160, 80$. Each layer was followed by a batch norm as well. We used the DSS implementation provided by Maron et al. (2020). As

mentioned earlier, we also trained a scaled DSS to match the number of parameters in the KAN-based models. We used a batch size of 64 as in Maron et al. (2020). For each configuration, we trained the models with 5 different seeds. We report in Table 3 the training durations for all models.

| Model | 60 | 120 | 600 | 900 | 1200 |
|---|---|---|---|---|---|
| DSS | $76.1 \pm 0.3$ | $93.9 \pm 0.6$ | $240.0 \pm 0.3$ | $356.7 \pm 25.9$ | $481.0 \pm 30.8$ |
| scaled DSS | $26.9 \pm 2.1$ | $32.8 \pm 1.6$ | $66.8 \pm 3.9$ | $125.2 \pm 68.4$ | $58.4 \pm 21.7$ |
| FS-KAN | $422.6 \pm 2.5$ | $499.7 \pm 5.1$ | $1183.7 \pm 10.9$ | $2025.4 \pm 29.4$ | $2228.9 \pm 315.2$ |
| Efficient FS-KAN | $300.8 \pm 6.0$ | $377.7 \pm 13.6$ | $1146.5 \pm 275.4$ | $1315.9 \pm 51.4$ | $1654.8 \pm 194.8$ |

(a) Low data regime

| Model | 6000 | 12000 | 18000 | 24000 | 30000 |
|---|---|---|---|---|---|
| DSS | $2e3 \pm 5.4$ | $3.9e3 \pm 7.6$ | $5.9e3 \pm 8.5$ | $7.8e3 \pm 12.6$ | $9.7e3 \pm 10.0$ |
| scaled DSS | $150.3 \pm 1.8$ | $385.4 \pm 147.6$ | $1.0e3 \pm 25.3$ | $1.0e3 \pm 437.1$ | $1.3e3 \pm 553.6$ |
| FS-KAN | $9.4e3 \pm 95.2$ | $1.9e4 \pm 164.5$ | $2.8e4 \pm 102.2$ | $3.8e4 \pm 245.9$ | $4.5e4 \pm 1098.8$ |
| Efficient FS-KAN | $7.8e3 \pm 79.8$ | $1.5e4 \pm 275.6$ | $2.3e4 \pm 219.2$ | $3.1e4 \pm 497.9$ | $3.8e4 \pm 827.0$ |

(b) High data regime

Table 3: Training time (seconds) of models for signal classification task (5). All models were trained for 200 epochs with a batch size of 64.

**Hyper-parameters.** We perform a grid search over training learning rates $\mu \in \{10^{-2}, 10^{-3}, \dots, 10^{-6}\}$ and regularization loss coefficients $\eta \in \{0, 10^{-2}, 10^{-4}\}$ with 6000 training samples. Hyper-parameters are selected based on the highest validation accuracy. All models achieve their best validation performance with $\mu = 10^{-3}$ and $\eta = 10^{-2}$.

## C.3   POINT CLOUDS CLASSIFICATION

**Data preparation.** For this experiment, we used ModelNet40 (Wu et al., 2015), which contains around $12 \times 10^3$ point clouds from different 40 classes. Each point cloud contains $n = 1024$ points represented by $d = 3$ spatial coordinates. We used the normalized and sampled point clouds by Qi et al. (2017) with no data augmentations. For the validation set, we used a $90\% - 10\%$ split of the full train set.

**Networks and training.** All invariant models are composed of equivariant layers, followed by an invariant layer and a standard output layer of width 40. FS-KAN and efficient KANs are composed of 2 equivariant FS-KA layers of width 36, each followed by a $1D$ batch norm. The invariant layer has an output dimension of 36 as well. The DeepSets model has the same architecture, only with a width of 128 to match the number of parameters ($5.6 \times 10^4$). The non-equivariant KAN consists of 3 hidden layers of width 16, but since it cannot handle varying input sizes, its parameter count depends on the number of input points, ranging from $4.2 \times 10^4$ to $5.0 \times 10^5$ parameters. The transformer model uses an embedding dimension of 38 with 3 layers ($5.5 \times 10^4$ parameters), implemented with an input projection layer, transformer encoder layers with multi-head attention and a classification head with intermediate dimensionality reduction. For training and testing, we used a batch size of 32. Additionally, we conducted an experiment to measure the average runtime and peak GPU memory usage for both the training and inference steps for the FS-KANs and DeepSets, as reported in Tables 4 and 5. We also conducted experiments in the high-data regime using the full ModelNet40 dataset (8,859 training samples with 1,024 points per cloud), a standard benchmark in 3D computer vision. The results, presented in Table 6, demonstrate that FS-KAN variants achieve comparable accuracy with established architectures like DeepSets and Point-Transformer in this setting, complementing our low-data regime analysis.

**Hyper-parameters.** We performed a grid search over training learning rates $\mu \in \{10^{-2}, 5 \times 10^{-3}, 10^{-3}, 5 \times 10^{-4}, 10^{-4}, 5 \times 10^{-5}\}$ and regularization loss coefficients $\eta \in \{10^{-5}, 10^{-3}, 0\}$. The training and validation sets consist of point clouds with $n = 1024$ points. The training set size for the search is 600. For FS-KAN models, the best validation performance is achieved with $\mu = 10^{-2}$ and $\eta = 10^{-5}$, while for DeepSets, the best configuration is $\mu = 10^{-2}$ and $\eta = 0$. The transformer model achieves optimal performance with $\mu = 10^{-3}$ and $\eta = 10^{-5}$, and the non-equivariant KAN with $\mu = 10^{-2}$ and $\eta = 0$.

(a) Train time per step (ms)

| Model | n=64 | n=128 | n=256 | n=512 | n=1024 |
|---|---|---|---|---|---|
| DeepSets | $4.09 \pm 0.46$ | $3.43 \pm 1.13$ | $4.22 \pm 0.65$ | $2.53 \pm 1.02$ | $1.97 \pm 0.24$ |
| Efficient FS-KAN | $6.19 \pm 1.50$ | $13.17 \pm 0.77$ | $6.11 \pm 0.07$ | $14.57 \pm 0.11$ | $41.70 \pm 0.35$ |
| FS-KAN | $6.33 \pm 0.14$ | $7.21 \pm 2.26$ | $7.53 \pm 1.62$ | $21.04 \pm 0.19$ | $61.78 \pm 0.53$ |
| Point-Transformer | $6.76 \pm 0.43$ | $6.56 \pm 0.09$ | $6.60 \pm 0.24$ | $17.67 \pm 6.76$ | $63.41 \pm 20.48$ |
| Standard KANs | $4.79 \pm 0.28$ | $5.41 \pm 1.53$ | $6.26 \pm 2.16$ | $4.84 \pm 0.36$ | $6.12 \pm 0.52$ |

(b) Inference time per step (ms)

| Model | n=64 | n=128 | n=256 | n=512 | n=1024 |
|---|---|---|---|---|---|
| DeepSets | $0.28 \pm 0.02$ | $0.29 \pm 0.03$ | $0.28 \pm 0.02$ | $0.27 \pm 0.01$ | $0.54 \pm 0.03$ |
| Efficient FS-KAN | $2.00 \pm 0.04$ | $2.00 \pm 0.06$ | $2.00 \pm 0.03$ | $4.94 \pm 0.03$ | $13.39 \pm 0.09$ |
| FS-KAN | $2.01 \pm 0.05$ | $2.04 \pm 0.04$ | $2.02 \pm 0.03$ | $7.11 \pm 0.03$ | $19.81 \pm 0.15$ |
| Point-Transformer | $0.61 \pm 0.02$ | $0.60 \pm 0.01$ | $0.60 \pm 0.01$ | $2.01 \pm 1.78$ | $6.54 \pm 6.47$ |
| Standard KANs | $1.20 \pm 0.01$ | $1.20 \pm 0.01$ | $1.22 \pm 0.00$ | $1.21 \pm 0.01$ | $1.19 \pm 0.01$ |

Table 4: Comparison of training times a and inference times b (ms) per step for different models on the point cloud classification task (batch size = 64) and for point clouds with $n$ points. We measure the runtime over 10 steps for each configuration and report the mean and standard deviation.

(a) Training phase (MB)

| Model | n=64 | n=128 | n=256 | n=512 | n=1024 |
|---|---|---|---|---|---|
| DeepSets | 31.23 | 45.28 | 73.37 | 129.56 | 241.94 |
| Efficient FS-KAN | 117.16 | 216.44 | 401.38 | 786.75 | 1556.00 |
| FS-KAN | 157.81 | 301.25 | 566.66 | 1120.84 | 2222.47 |
| Point-Transformer | 70.01 | 148.90 | 397.59 | 1323.53 | 4797.90 |
| Standard KANs | 18.78 | 21.16 | 25.99 | 35.46 | 54.53 |

(b) Inference phase (MB)

| Model | n=64 | n=128 | n=256 | n=512 | n=1024 |
|---|---|---|---|---|---|
| DeepSets | 23.20 | 29.24 | 41.34 | 65.53 | 145.90 |
| Efficient FS-KAN | 43.05 | 68.36 | 114.71 | 215.14 | 406.52 |
| FS-KAN | 42.57 | 68.86 | 116.33 | 219.02 | 414.89 |
| Point-Transformer | 22.95 | 31.50 | 61.22 | 169.28 | 577.40 |
| Standard KANs | 18.89 | 21.35 | 26.31 | 36.07 | 55.70 |

Table 5: Comparison of GPU memory usage during training a and inference b (MB) for different models on the point cloud classification task and for point clouds with $n$ points.

| Model | DeepSets | FS-KAN | Efficient FS-KAN | Point-Transformer | Standard KAN |
|---|---|---|---|---|---|
| Test Accuracy | $0.844 \pm 0.005$ | $0.837 \pm 0.007$ | $0.840 \pm 0.005$ | $0.840 \pm 0.008$ | $0.232 \pm 0.074$ |

Table 6: Test accuracy on full ModelNet40 dataset (high-data regime)

## C.4 CONTINUAL LEARNING ON POINT CLOUDS

**Data preparation.** For task A, we followed the procedure described in Appx. C.3. For task B, we applied geometric transformations to the task A dataset consisting of: (i) random scaling sampled uniformly from $[0.95, 1.0]$, (ii) 3D rotation via axis-angle representation with rotation angles drawn from $\mathcal{N}(0, 0.5^2)$, and (iii) random translation with components sampled from $\mathcal{N}(0, 0.1^2 \mathbf{I})$.

**Network and training.** We use the same models from Appx. C.3. We employ identical hyperparameters for each model across both tasks. Each phase consisted of 1000 epochs, with the optimizer reinitialized at the beginning of each phase.

### C.5 SEMI-SUPERVISED RATING PREDICTION EXPERIMENT

**Data preparation.** We used the MovieLens 100k dataset (Harper & Konstan, 2015) along with smaller versions of Douban, Flixster, and Yahoo, as introduced by Monti et al. (2017). In the Flixster dataset, where ratings are given in 0.5 increments, we labeled rating 0.5 as 1 represented non-integer ratings as probabilistic mixtures (e.g., a rating of 3.5 was modeled as a 50% chance of 3 and 50% chance of 4). For the Yahoo dataset, ratings ranging from 1 to 100 were clustered into 5 categories and treated as $1 - 5$ values.

To simulate low-data scenarios, we uniformly sampled subsets of rows and columns from each rating matrix. Within each resulting sub-matrix, we further sub-sampled the observed entries to reduce training density. The train set was composed of uniformly sampled sub-matrices of the sparse matrix. The train set consists of 32 sub-matrices, and the test and validation sets have 64 samples each. The full data sampling process is illustrated in figure 8. Data was split into train, validation, and test sets using the U1 split defined by Harper & Konstan (2015).

**Network and training.** All models are composed of 9 equivariant layers, each followed by a skip connection and batch norm. KAN and FS-KAN architectures used layers of width 16 while SSEM uses layers of width 256 as detailed on Hartford et al. (2018). A smaller SSEM with a width of 52 was also trained to match the parameter count of the KAN models. For each configuration, we trained the models with 5 different seeds.

**Hyper-parameter search** We perform a grid search over training learning rates $\mu \in \{10^{-2}, 10^{-3}, 10^{-4}, 10^{-5}\}$ and regularization loss coefficients $\eta \in \{10^{-3}, 10^{-5}, 0\}$, using approximately 600 ratings per run. Hyper-parameters are selected based on the lowest validation RMSE. The best-performing configurations are as follows:

- **KAN** and **Efficient FS-KAN**: $\mu = 10^{-4}$, $\eta = 10^{-5}$ across all datasets.
- **SSEM** and **Scaled SSEM**:
  - On ML-100K, Flixster, and Yahoo: $\mu = 10^{-4}$, $\eta = 10^{-5}$.
  - On Douban:
    * SSEM: $\mu = 10^{-3}$, $\eta = 10^{-5}$.
    * Scaled SSEM: $\mu = 10^{-3}$, $\eta = 0$.

## D  HYPERPARAMETER STUDY ON FS-KAN MODELS

We evaluated our FS-KAN model on point cloud classification (section 5), ablating over different architectural configurations. Specifically, we examined different widths $w \in \{36, 64, 128\}$, depths $L \in \{2, 4, 6\}$, and grid sizes $g \in \{3, 4, 5, 6\}$ of the spline functions. We note that increasing each of these parameters increases the model size. For each hyper-parameter configuration, we trained models on 600 samples, each containing $n = 256$ points.

**Results.** Results are presented in Table 7. Our findings demonstrate that models with $d = 4$ consistently achieve superior test accuracies across all examined grid sizes, suggesting an optimal architectural configuration at moderate depth levels. Increasing the layer width from 36 to 128 yields notable performance improvements (approximately $0.010 - 0.015$), with width 128 achieving the best results. However, this comes at a significant computational cost: as shown in Tables 8 and 10, training time and memory usage scale super-linearly with width and more linearly with depth, while grid size increases show moderate impact. Regarding grid size, larger values ($g = 4$ to $g = 6$) generally provide better accuracy with diminishing returns. Considering both accuracy (Table 7) and computational efficiency (Tables 8, 9, 10), our results indicate that a configuration of $d = 4, w = 128, g = 4$ optimally balances classification performance and model complexity for point cloud classification tasks.

## E  LARGE LANGUAGE MODEL (LLM) USAGE

We used LLMs to assist with writing and polishing portions of this paper, including improving clarity of technical explanations, refining grammar and flow, and enhancing overall readability. All

| Depth | Width | $g = 3$ | $g = 4$ | $g = 5$ | $g = 6$ |
|---|---|---|---|---|---|
| | 36 | 0.637 ± 0.008 | 0.637 ± 0.008 | 0.644 ± 0.003 | 0.640 ± 0.009 |
| 2 | 64 | 0.644 ± 0.012 | 0.644 ± 0.007 | 0.650 ± 0.004 | 0.653 ± 0.006 |
| | 128 | 0.646 ± 0.008 | 0.648 ± 0.004 | 0.653 ± 0.009 | 0.652 ± 0.009 |
| | 36 | 0.645 ± 0.011 | 0.645 ± 0.009 | 0.643 ± 0.010 | 0.647 ± 0.005 |
| 4 | 64 | 0.644 ± 0.004 | 0.653 ± 0.007 | 0.652 ± 0.007 | 0.650 ± 0.007 |
| | 128 | 0.649 ± 0.006 | 0.658 ± 0.007 | 0.654 ± 0.011 | 0.660 ± 0.002 |
| | 36 | 0.631 ± 0.010 | 0.637 ± 0.006 | 0.642 ± 0.013 | 0.641 ± 0.005 |
| 6 | 64 | 0.642 ± 0.005 | 0.643 ± 0.009 | 0.640 ± 0.009 | 0.647 ± 0.005 |
| | 128 | 0.639 ± 0.007 | 0.635 ± 0.010 | 0.639 ± 0.011 | 0.640 ± 0.008 |

Table 7: FS-KAN test accuracy results for different architectural configurations on point cloud classification

Table 8: Average Training Time (ms) for Point Cloud Size n=256

| Depth | Width | $g = 3$ | $g = 4$ | $g = 5$ | $g = 6$ |
|---|---|---|---|---|---|
| | 36 | 8.197 ± 1.576 | 8.487 ± 1.497 | 8.545 ± 1.571 | 7.802 ± 1.295 |
| 2 | 64 | 9.132 ± 1.465 | 9.778 ± 2.835 | 9.901 ± 2.860 | 9.655 ± 0.994 |
| | 128 | 11.313 ± 0.037 | 13.609 ± 0.019 | 15.928 ± 0.143 | 18.320 ± 0.053 |
| | 36 | 14.412 ± 2.561 | 15.252 ± 2.329 | 14.788 ± 2.177 | 13.553 ± 2.556 |
| 4 | 64 | 13.234 ± 2.637 | 15.456 ± 4.376 | 14.750 ± 2.764 | 16.762 ± 2.579 |
| | 128 | 28.023 ± 0.184 | 33.668 ± 0.247 | 39.656 ± 0.163 | 45.981 ± 0.301 |
| | 36 | 18.637 ± 3.770 | 18.774 ± 3.310 | 16.998 ± 1.292 | 16.984 ± 0.537 |
| 6 | 64 | 17.848 ± 0.956 | 19.603 ± 4.190 | 19.048 ± 0.050 | 23.211 ± 0.104 |
| | 128 | 44.329 ± 0.257 | 54.041 ± 0.241 | 63.376 ± 0.232 | 73.487 ± 0.292 |

Table 9: Average Inference Time (ms) for Point Cloud Size n=256

| Depth | Width | $g = 3$ | $g = 4$ | $g = 5$ | $g = 6$ |
|---|---|---|---|---|---|
| | 36 | 2.002 ± 0.024 | 2.039 ± 0.019 | 2.002 ± 0.034 | 2.177 ± 0.298 |
| 2 | 64 | 1.989 ± 0.033 | 2.249 ± 0.549 | 2.375 ± 0.801 | 2.057 ± 0.128 |
| | 128 | 3.821 ± 0.013 | 4.853 ± 0.004 | 5.916 ± 0.003 | 7.040 ± 0.004 |
| | 36 | 3.449 ± 0.257 | 3.461 ± 0.295 | 3.335 ± 0.090 | 3.442 ± 0.076 |
| 4 | 64 | 3.348 ± 0.083 | 3.536 ± 0.482 | 3.487 ± 0.417 | 3.425 ± 0.196 |
| | 128 | 8.066 ± 0.096 | 10.448 ± 0.063 | 13.149 ± 0.073 | 15.946 ± 0.113 |
| | 36 | 4.808 ± 0.225 | 4.640 ± 0.064 | 4.700 ± 0.094 | 4.697 ± 0.117 |
| 6 | 64 | 4.630 ± 0.054 | 4.879 ± 0.303 | 5.040 ± 0.223 | 5.574 ± 0.144 |
| | 128 | 12.838 ± 0.022 | 16.699 ± 0.012 | 20.867 ± 0.037 | 25.038 ± 0.028 |

research contributions, experimental design, analysis, and conclusions are entirely our own work. The LLM was used solely as a writing assistance tool to improve quality.

Table 10: Peak Training GPU Memory Usage (MB) for Point Cloud Size n=256

| Depth | Width | $g=3$ | $g=4$ | $g=5$ | $g=6$ |
|---|---|---|---|---|---|
| | 36 | 234.305 | 269.117 | 301.245 | 330.756 |
| 2 | 64 | 399.340 | 453.631 | 504.164 | 556.456 |
| | 128 | 781.065 | 885.992 | 990.793 | 1096.722 |
| | 36 | 498.823 | 575.209 | 646.656 | 710.748 |
| 4 | 64 | 865.529 | 984.267 | 1099.116 | 1215.730 |
| | 128 | 1718.178 | 1952.368 | 2186.433 | 2421.625 |
| | 36 | 763.967 | 882.019 | 994.067 | 1091.333 |
| 6 | 64 | 1331.842 | 1514.778 | 1694.068 | 1875.005 |
| | 128 | 2655.290 | 3018.744 | 3382.073 | 3746.528 |

Table 11: Peak Inference GPU Memory Usage (MB) for Point Cloud Size n=256

| Depth | Width | $g=3$ | $g=4$ | $g=5$ | $g=6$ |
|---|---|---|---|---|---|
| | 36 | 58.441 | 63.805 | 68.857 | 72.668 |
| 2 | 64 | 89.884 | 98.951 | 106.262 | 114.455 |
| | 128 | 166.093 | 181.824 | 198.429 | 215.912 |
| | 36 | 59.138 | 63.706 | 69.743 | 72.734 |
| 4 | 64 | 91.905 | 101.224 | 108.784 | 117.229 |
| | 128 | 174.123 | 190.856 | 208.463 | 226.948 |
| | 36 | 59.771 | 64.389 | 70.504 | 73.643 |
| 6 | 64 | 93.925 | 103.496 | 111.307 | 120.004 |
| | 128 | 182.153 | 199.888 | 218.498 | 237.984 |

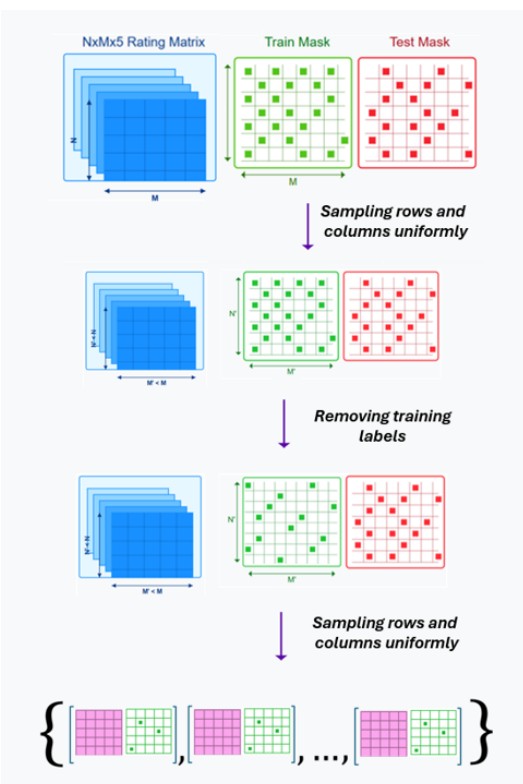

Figure 8: Rating matrix sub-sampling pipeline. We uniformly sample users and items, followed by removing observed entries to simulate extreme data scarcity. Finally, we repeatedly sample rows and columns to generate multiple sub-matrices used as training and test batches.

