# OpenReview forum: "FS-KAN: Permutation Equivariant Kolmogorov-Arnold Networks via Function Sharing"
_ICLR.cc/2026/Conference — ICLR 2026 Poster_

### Official Review · Reviewer_G8oo · 2025-10-31

**Soundness:** 3
**Presentation:** 3
**Contribution:** 3
**Rating:** 4
**Confidence:** 3

**Summary:**

This paper proposes FS-KAN, a framework that enforces permutation equivariance and invariance in Kolmogorov–Arnold Networks by sharing functions rather than weights. The authors establish an expressivity equivalence with parameter-sharing ReLU MLPs, showing that FS-KANs match their expressivity and thus inherit known universality guarantees for symmetry aware models. Empirically, across multiple symmetry tasks, FS-KANs outperform other baselines in low-data regimes, delivering better data efficiency while retaining KANs interpretability.

**Strengths:**

- The paper proves that FS-KAN captures all G-equivariant/invariant KA layers via function sharing. The proof is easy to follow with standard techniques in equivariant/invariant layer design.
- Emperical evident suggests that their FS-KAN is better comparing to other baselines in low data regime.

**Weaknesses:**

- In the point-cloud task (e.g., at ~1k training samples), other baseline match or surpass FS-KAN, suggesting its benefit diminishes as training size grows. This reduces the practical appeal of FS-KAN beyond low data settings.
- The computational runtime/memory analysis from Appendix C shows that FS-KAN incurs substantially higher runtime and memory usage than the other baselines.

**Questions:**

- Does function sharing over-constrain FS-KAN? With many spline functions tied across positions (Appendix Fig. 7), could the sharing pattern overly limit capacity and hinder learning in practice?
- The appendix indicates substantially higher runtime and memory than other baselines. Can runtime/memory for FS-KAN be improved?

---

> ### Author Response · Authors · 2025-11-20
>
> We sincerely thank the reviewer for their evaluation and feedback. We are pleased that the reviewer recognized that **"empirical evidence suggests that their FS-KAN is better compared to other baselines"** and that **"the paper proves that FS-KAN captures all G-equivariant/invariant KA layers via function sharing."** We address the reviewer's concerns below:
>
> 1. **"In the point-cloud task (e.g., at ~1k training samples), other baseline match or surpass FS-KAN, suggesting its benefit diminishes as training size grows. This reduces the practical appeal of FS-KAN beyond low data settings"**
> We acknowledge that FS-KAN's advantages are most pronounced in low-data regimes, which is consistent with our central claims. However, we respectfully note that low-data settings are extremely common across many fields that utilize ML tools, including scientific domains, medical applications, specialized industrial tasks, and scenarios with expensive data collection. In these contexts, practitioners often face severe data limitations where even modest accuracy improvements can have significant practical value. For such applications, the trade-off of higher computational cost for better performance in data-scarce settings is often acceptable and even preferable. Our work demonstrates that FS-KANs provide a viable solution precisely for these important use cases where data efficiency is paramount. While we agree that standard parameter-sharing MLPs become more competitive as data abundance increases, this does not diminish the practical appeal of FS-KANs—rather, it clarifies their appropriate deployment context: situations where data is limited and accuracy gains justify additional computational expense.
>
> 2. **"Does function sharing over-constrain FS-KAN? With many spline functions tied across positions (Appendix Fig. 7), could the sharing pattern overly limit capacity and hinder learning in practice?"**
> Thank you for this important question. As we prove in our paper, if we require each KA layer in the model to be equivariant, then restricting it to function sharing does not over-constrain FS-KAN (Propositions 4 and 6 establish that our function-sharing parameterization fully characterizes the space of equivariant KA layers). Thus, the function-sharing "constraint" should be viewed not as a limitation, but as an architectural design choice that appropriately matches the symmetry structure of the data. Nonetheless, we show that FS-KA layers have the same expressivity as parameter-sharing layers (Propositions 6-7), and therefore can be limited for some specific group representations such as in graph learning [1].
>
>
> We hope our responses have adequately addressed the reviewer's concerns and clarified the contributions of our work. Given the revisions we have made to strengthen the manuscript based on this feedback, we would be grateful if the reviewer would consider raising their score to a positive one.
>
> [1] Maron, Haggai, et al. "Provably powerful graph networks." Advances in neural information processing systems 32 (2019).‏

---

> > ### Comment · Reviewer_G8oo · 2025-11-27
> > **Official Comment by Reviewer**
> >
> > I thank the authors for their response. I believe my concerns have been satisfactorily addressed; however, there remain some limitations, as FS-KAN does not demonstrate substantial benefits in the large-data regime. I view this as inherent to the model itself rather than a consequence of the rigor of the theoretical analysis. Accordingly, I am increasing my score to 6.

---

### Official Review · Reviewer_DxdS · 2025-11-01

**Soundness:** 3
**Presentation:** 3
**Contribution:** 1
**Rating:** 2
**Confidence:** 3

**Summary:**

The paper proposes FS-KAN and function sharing, enabling Kolmogorov–Arnold Networks (KAN) to implement group equivariance. This design unifies parameter sharing and functions sharing enabling transfer of known results (e.g., universality). Experiments show competitive performance, especially in low-data regimes.

**Strengths:**

- Well written and clearly structured with working examples.
- Proposition 2 is a rigorous result which requires new and different proofs from the ones known in the weight sharing literature for equivariant neural networks.
- The authors propose a perspective that unifies neural networks and KANs. Function sharing for KANs cleanly recovers common equivariant designs for permutation representations of groups.

**Weaknesses:**

- **Insufficient novelty.** The introduction of the concept of function sharing is a simple adaptation of the regular concept of weight sharing for equivariant neural networks, as presented in [1]. The majority of main theoretical statements except for Proposition 2 largely transfer as it is from the weight sharing literature for equivariant deep learning.
- **Missing analysis of low-frequency learning and spectral bias.** This topic is currently central to KAN research [2]. In particular, KANs are known to avoid the low-frequency bias of standard neural networks, enabling them to perform better on high-frequency targets (e.g., certain PDE solutions). These properties make KANs especially relevant for AI-for-Science; yet the manuscript does not investigate this direction, even though enforcing equivariance to the symmetries of specific PDEs could potentially improve performance.

[1] S. Ravanbakhsh, *Equivariance Through Parameter-Sharing*, 2017 \
[2] Y. Wang et al., *On the expressiveness and spectral bias of KANs*, 2024

**Questions:**

**Interpretability claim.**
One of KANs’ most advertised features is improved interpretability, and FS-KANs build on this. However, this claim is contested: learnable activations are not identifiable up to scale and permutations, which makes post-hoc interpretation fragile at the local scale. At larger scales, even the KAN authors acknowledge that “a KAN may only remain interpretable when the network scale is relatively small” (p. 21 of [3]).
To what extent does enforcing equivariance mitigate this fragility?

[3] Z. Liu, "KAN 2.0: Kolmogorov-Arnold Networks Meet Science", 2025

---

> ### Author Response · Authors · 2025-11-20
>
> We sincerely thank the reviewer for their detailed feedback and evaluation. We are pleased that the reviewer found our paper **"well written and clearly structured with working examples,"** recognized that **"Proposition 2 is a rigorous result which requires new and different proofs from the ones known in the weight sharing literature,"** and appreciated that **"the authors propose a perspective that unifies neural networks and KANs"** where **"function sharing for KANs cleanly recovers common equivariant designs."** We address the reviewer's concerns below:
> 1. **"Insufficient novelty. The introduction of the concept of function sharing is a simple adaptation of the regular concept of weight sharing for equivariant neural networks"**
> We respectfully disagree with this assessment. Although our work draws on established principles of weight sharing in equivariant networks, adapting these concepts to function sharing in KANs involves substantial theoretical and practical challenges. Our work addresses a clear gap in the literature by providing the first systematic treatment of symmetries in KAN architectures. The key contributions include:
> a. We demonstrated that equivariant KA layers admit multiple representations (e.g., Equation 3), and proved that every such layer possesses an equivalent function-sharing form (Propositions 2, 4). Establishing these results required non-trivial arguments involving orbit decompositions and careful mathematical treatment.
>
> b. We extended the KAN framework to accommodate multiple feature channels and higher-order tensors (Proposition 5, Section 3.3), which necessitated rigorous analysis to handle the added complexity. While our proofs draw upon established algebraic tools from equivariant deep learning, this does not diminish their significance—it shows how these principles must be thoughtfully adapted to the function-based parameterization that distinguishes KANs from traditional architectures.
>
> c. We prove that FS-KANs and parameter-sharing MLPs have equivalent expressive power (Propositions 6-7). This equivalence is significant because it allows us to transfer well-established expressivity results from the equivariant MLP literature to FS-KANs (Corollary 4.1), including universality theorems for set functions and WL-test characterizations for graph networks. This connection provides both fundamental limitations and guarantees for FS-KAN expressivity that were not previously established.
>
> d. We provided comprehensive experimental validation across diverse data symmetries, addressing a deficiency in the existing KAN literature. The integration of theoretical foundations with extensive empirical investigation represents a substantial contribution toward understanding how KANs can effectively leverage structural symmetries in data.
>
>
> 2. **"Missing analysis of low-frequency learning and spectral bias. This topic is currently central to KAN research. In particular, KANs are known to avoid the low-frequency bias of standard neural networks, enabling them to perform better on high-frequency targets (e.g., certain PDE solutions)"**
> We appreciate the reviewer raising this important topic. While spectral bias and high-frequency learning are indeed central to KAN research, we are not aware of established benchmark datasets or concrete experimental setups that would allow us to systematically evaluate these properties in the context of permutation-equivariant architectures. The existing literature on KANs' spectral advantages primarily focuses on PDE solving tasks, which typically involve continuous spatial domains rather than the discrete permutation-symmetric data types (sets, graphs, user-item interactions) that our framework addresses. The connection between spectral bias, symmetry constraints, and permutation-equivariant learning remains an open question that would require careful design of appropriate tasks and evaluation protocols. We would be happy to explore this direction if the reviewer could suggest specific datasets or experimental setups where the interplay between function sharing, permutation equivariance, and spectral properties could be meaningfully evaluated. We acknowledge this as valuable future work and believe establishing the foundational framework for permutation-equivariant KANs is an important first step toward such investigations.

---

> ### Author Response · Authors · 2025-11-20
>
> 3. **"One of KANs' most advertised features is improved interpretability, and FS-KANs build on this. However, this claim is contested... To what extent does enforcing equivariance mitigate this fragility?"**
> Enforcing equivariance through function sharing directly mitigates interpretability fragility. FS-KAN constraints force many functions to be identical according to the group action, acting as a strong regularizer that helps converge to more stable and meaningful representations. This dramatically reduces the number of functions requiring interpretation—for example, our $S_n$-FS-KAN has only 2 shared functions per equivariant layer instead of $n^2$ independent functions. As shown in Figures 3 and 7, this makes the learned structure immediately apparent: for example, in Fig. 3 , shared quadratic-like splines in equivariant layers and exponential behavior in invariant layers clearly reflect the underlying mathematical structure, enabling interpretation of each layer and potentially scaling to larger networks.
>
>
> We believe our responses address the reviewer's concerns and demonstrate the non-trivial nature and significance of our theoretical and empirical contributions. In light of these clarifications and the revisions we have made to the manuscript, we respectfully ask the reviewer to reconsider their evaluation and hope they will find our work merits a higher score.

---

### Official Review · Reviewer_mRDw · 2025-11-01

**Soundness:** 3
**Presentation:** 3
**Contribution:** 3
**Rating:** 6
**Confidence:** 2

**Summary:**

This paper introduces FS-KAN, a principled framework for building permutation-equivariant and invariant neural networks by extending parameter-sharing schemes to function sharing. The approach unifies and generalizes prior work on equivariant KANs, providing a theoretical proof that FS-KANs have equivalent expressive power to traditional parameter-sharing MLPs. Empirically, FS-KANs achieve strong performance and data efficiency across diverse symmetry-aware tasks, including signal classification, point cloud recognition, and matrix completion, particularly in low-data regimes. The architecture retains the interpretability and adaptability of KANs while significantly improving generalization for data with built-in symmetries

**Strengths:**

- The paper presents a theoretically sound and general framework for permutation-equivariant KANs, extending the idea of parameter sharing to function-level sharing. It also provides clear mathematical proofs showing expressivity equivalence with parameter-sharing MLPs.
- The proposed architecture maintains the interpretability of KANs, as demonstrated in visual examples where symmetric function structures emerge naturally.
- The efficient FS-KAN variant reduces computational and memory costs while preserving equivariance, making the method more practical.

**Weaknesses:**

- Despite the good performance, computational cost remains high, particularly compared to standard MLP-based models in high-data regimes.
- The empirical analysis, while broad, relies mostly on synthetic or small-scale datasets, leaving scalability to large real-world systems untested.
- The practical interpretability claims are qualitative; the paper lacks systematic metrics quantifying how function sharing aids transparency.

**Questions:**

- How does FS-KAN training scale computationally and memory-wise with input dimension and group size, compared to DeepSets or GNN baselines?
- Can FS-KANs be adapted for continuous or mixed-group symmetries (e.g., rotations, reflections) beyond discrete permutations?

---

> ### Author Response · Authors · 2025-11-20
>
> We sincerely thank the reviewer for their positive evaluation and thoughtful feedback. We are pleased that the reviewer recognized that **"the paper presents a theoretically sound and general framework for permutation-equivariant KANs," **appreciated that **"the proposed architecture maintains the interpretability of KANs, as demonstrated in visual examples,"** and acknowledged that **"the efficient FS-KAN variant reduces computational and memory costs while preserving equivariance."** We address the reviewer's concerns below:
>
> 1.** "Despite the good performance, computational cost remains high"**
> We appreciate the reviewer's concerns regarding computational efficiency. We acknowledge that KAN-based models currently face computational challenges; however, improving KAN runtime is an active area of research in the broader KAN literature, with ongoing efforts to develop more efficient variants proceeding in parallel with explorations of KANs' applicability to new domains. Critically, our function-sharing framework will directly benefit from any future advances in KAN efficiency, as the architectural contribution—the function-sharing mechanism for permutation equivariance—is orthogonal to the specific KAN parameterization used. Any improvements in KAN implementations (better spline methods, alternative basis functions, etc.) can be seamlessly integrated into FS-KAN with minimal modification. While we agree the current accuracy-efficiency trade-off limits immediate widespread deployment, we view this work as foundational research that establishes the viability of KAN-based permutation-equivariant architectures, which will become increasingly practical as the community develops more efficient KAN variants.
> Importantly, in low-data regimes—which are prevalent in medical applications and specific industrial tasks where data collection is expensive or limited—where FS-KAN excels, the performance gains are substantial: for example, with 600 training samples in signal classification, FS-KAN outperforms scaled DSS by approximately 10% in accuracy (Figure 4a), and with 200 visible ratings in the Douban dataset, FS-KAN achieves about 20% lower RMSE compared to SSEM (Figure 6c). This performance improvement in low-data regimes provides practitioners with a practical advantage that could justify the trade-off.
>
> 2. **"The empirical analysis, while broad, relies mostly on synthetic or small-scale datasets, leaving scalability to large real-world systems untested"**
> To address the reviewer's concerns regarding scalability and large-scale benchmarks, we conducted point cloud classification experiments using the complete ModelNet40 dataset—a widely-used benchmark in 3D computer vision comprising 8,859 training samples with 1,024 points per cloud. provide insight into FS-KAN behavior in high-data settings, complementing the low-data analyses featured in our manuscript.
>
> | Model          |   DeepSets    |    FS-KAN     | Efficient FS-KAN | Point-Transformer | Standard KAN  |
> |:---------------|:-------------:|:-------------:|:----------------:|:-----------------:|:-------------:|
> | Test Accuracy  | 0.844 ± 0.005 | 0.837 ± 0.007 | 0.840 ± 0.005    | 0.840 ± 0.008     | 0.232 ± 0.074 |
>
> Our findings show that FS-KAN variants deliver accuracy comparable to well-established methods such as DeepSets and Point-Transformer when data is abundant. Nevertheless, the substantially elevated computational demands render KAN-based models less practical for high-data scenarios, mirroring our observations from the signal classification experiments. It is worth noting that active research into KAN architectures, including investigations of alternative activation functions and optimized implementations, along with potential refinements to the FS-KAN framework through advanced optimization and regularization techniques, may improve this efficiency-performance balance over time. We remain open to evaluating FS-KAN on additional large-scale datasets if the reviewer considers that this would provide valuable additional evidence.

---

> > ### Author Response · Authors · 2025-11-20
> >
> > 3. **"The practical interpretability claims are qualitative; the paper lacks systematic metrics quantifying how function sharing aids transparency"**
> > We appreciate this feedback. We have made our best effort to demonstrate interpretability through multiple qualitative examples (Figures 3, 7, and additional examples in Appendix C.1), showing how function sharing makes the learned symmetry structure visually explicit and simplifies the network representation. However, we acknowledge that quantifying interpretability remains a challenging open problem in the field. To the best of our knowledge, there are no established metrics for systematically quantifying the interpretability benefits of function sharing in the context of symmetric neural architectures. If the reviewer is aware of specific metrics or evaluation protocols that would be appropriate for this setup, we would be happy to include such quantitative analysis in a revised version. We believe our qualitative demonstrations still provide meaningful evidence of enhanced interpretability, but we are open to exploring additional ways to substantiate these claims if concrete suggestions are provided.
> > 4. **"How does FS-KAN training scale computationally and memory-wise with input dimension and group size, compared to DeepSets or GNN baselines?"**
> > Thank you for this question. We clarify that the group action for a given model architecture is fixed, as the group defines the parameter-sharing scheme. However, certain architectures can accommodate multiple group sizes within the same sharing pattern—for example, DeepSets uses the symmetric group structure, and has the same layer formulation for $S_n,  for different values of n, by processing varying numbers of elements. Similarly, $S_n$-FS-KANs, like DeepSets and GNNs, can handle varying input sizes within the same sharing scheme, and the computational and memory scaling depends only on the input dimension, not the group size itself. We report comprehensive time and memory comparisons for different input sizes in Tables 3, 4. As shown in these tables, while FS-KAN variants require more computational time and memory than MLP-based architecture, the efficient FS-KAN variant provides considerable speedups and memory reductions compared to standard FS-KAN, demonstrating the practical benefits of our efficient layer design.
> > 5. **"Can FS-KANs be adapted for continuous or mixed-group symmetries (e.g., rotations, reflections) beyond discrete permutations?"**
> > Thank you for this question. Extending to continuous groups would require fundamentally different approaches. Unlike permutation groups which admit closed-form discrete sharing patterns, continuous groups lack such closed-form solutions for the function-sharing structure. This would necessitate numerical methods to solve for equivariant constraints, significantly increasing computational complexity, or other more complex directions. We focus on permutation symmetries due to their efficient closed-form solutions and prevalence across important data types (sets, graphs, hierarchical structures). Extensions to continuous groups is valuable future work beyond our current scope.
> >
> > We sincerely appreciate the reviewer's positive evaluation of our work. We hope our detailed responses and the improvements made to the manuscript further strengthen the contribution, and we would be grateful if the reviewer would consider adjusting their score accordingly.

---

### Official Review · Reviewer_9HVo · 2025-11-01

**Soundness:** 3
**Presentation:** 2
**Contribution:** 2
**Rating:** 4
**Confidence:** 3

**Summary:**

The paper introduces a new approach to realize equivariant (or invariant) KANs. The underlying mechanism is a direct generalization of the weight-sharing used for MLPs, where now it becomes a function sharing of the learnable KAN functional components.
The work claims that the approach generalizes existing equivariant KAN architectures, and that the function-sharing approach is general, since any equivariant KA layer can be exactly represented by a FS-KA layer.
The authors then introduce an efficient version of the layer, provide examples intended to demonstrate how FS-KAN can be designed and how they generalize existing architectures, and prove that MLPs and FS-KANs can approximate each other to arbitrary accuracy. This fact is then used to derive universality properties of FS-KANs.
The experimental section convincingly demonstrate the potential of the new architecture.

**Strengths:**

- The related work clearly highlights the motivation of the work (simplification of existing approaches, variable size inputs, general permutation groups).
- The topic of combining KANs with equivariant modelling is of current interest, and the results cover both efficient implementation and analytical properties.
- The experimental section is clearly structured and it addresses a clearly defined set of questions.

**Weaknesses:**

- Efficient FS-KA layers: This seems to be a central novelty of the paper, as it enables applications and scaling to realistic problems. However, their definition is mainly textual and hard to process. I suggest to revise the paragraph "Efficient FS-KA layers" and make their definition more explicit.

- Generalization of the existing literature: There is an emphasis in Sections 1-2 on the fact that FS-KANs generalize and unify existing approaches. This fact is however not sufficiently demonstrated in the paper. Section 3.3 in particular has a "FS-KANs generalize previous equivariant KANs" paragraph, which only mentions some similarity with the existing literature.

- Theoretical analysis: Proposition 7 is formulated in an imprecise manner, as there is no dependency on the target accuracy $\varepsilon$, while clearly the MLP $f$ is in fact $f(\varepsilon)$. This becomes evident by inspecting the proof, but it should made clear upfront.
Especially, the width $d_{\max}$ of the MLP $f$ seems to be unboundedly growing in $\varepsilon$, since the construction is based on the 1989 Hornik et al. result.
Apart from updating the statement of Proposition 7, I also recommend to update the following sentences:  (I) "Both proofs provide constructive methods for building the corresponding networks",  following Proposition 7, which is not correct as it stands; (ii) "we show that for a specific permutation group G, both model families have the same expressive power in the uniform approximation sense", should rather state the the two families are dense in the same set of functions.

- The experimental results compare FS-KANs only with MLPs with shared parameters. Comparisons with other equivariant architectures should be considered.

**Questions:**

Apart from the points discussed above, there are the following minor points:

- The $\star$ notation in (1) is standard in KANs, but should be briefly introduced.

- Paragraph "Invariance, equivariance, and standard parameter-sharing": The problem is formulated for $V=V'=R^n$, instead of general $V=R^n$, $V'=R^{n'}$. It is unclear at this stage if this is a necessary assumption for the definition of FS-KA layers, or if the assumption is just for the sake of simplifying the presentation. Please clarify this.

- Equation (3) gives the wrong impression that only constant, zero-sum offsets are allowed. While this is clear from the context, I suggest modifying the example in (3) by using a functional offset.

- Proof of Proposition 6: The notions of grid size and $B$-splines ("grid size $G = 2$ with degree $k$ B-splines") are nowhere defined before.

---

> ### Author Response · Authors · 2025-11-20
>
> We sincerely thank the reviewer for their constructive feedback and thoughtful evaluation. We are pleased that the reviewer found **"the related work clearly highlights the motivation of the work,"** appreciated that **"the topic of combining KANs with equivariant modelling is of current interest,"** and recognized that **"the experimental section is clearly structured and addresses a clearly defined set of questions."** We address the reviewer's concerns below:
>
> 1. **"I suggest to revise the paragraph 'Efficient FS-KA layers'"**
> We have revised this paragraph to provide clearer exposition. Specifically, we now explicitly state that "For arbitrary permutation groups $G$, efficient FS-KAN layers are derived following the same principle: commuting element aggregations (sum or mean pooling) with application of shared functions according to the original sharing structure. In general, the efficiency of the layer is determined by the group structure." We discuss the efficiency and provide a more formal construction of efficient FS-KA layers for general permutation symmetry groups in Appendix A.3.
>
> 2. **"Section 3.3 in particular has a 'FS-KANs generalize previous equivariant KANs' paragraph, which only mentions some similarity with the existing literature"**
> We appreciate this observation and would like to clarify our contributions relative to existing work. The similarity we describe is not merely superficial—our framework exactly recovers previous equivariant KANs as special cases. Specifically, our $S_n$-FS-KAN (Equation 5) is mathematically equivalent to PointNet-KAN (Kashefi, 2024), and our $C_n$-FS-KAN is equivalent to KA-convolution networks (Bodner et al., 2024).. Our contribution lies in providing a general, principled framework that unifies these previously disparate approaches and extends to numerous other permutation symmetry groups not yet explored in the KAN literature (e.g., direct-product symmetries $S_n \times S_m$ for user-item interactions, hierarchical structures, multi-set interactions, and high-order relational data). This unification not only provides theoretical grounding for existing methods but also offers a systematic blueprint for designing equivariant KAN architectures for new symmetry groups.
>
> 3. **"Theoretical analysis: Proposition 7 is formulated in an imprecise manner, as there is no dependency on the target accuracy $\epsilon$... I also recommend to update the following sentences: (i) 'Both proofs provide constructive methods for building the corresponding networks', following Proposition 7, which is not correct as it stands; (ii) 'we show that for a specific permutation group G, both model families have the same expressive power in the uniform approximation sense', should rather state the two families are dense in the same set of functions."**
> We appreciate these notes and will update this section in our revision accordingly. Specifically, we will: (i) explicitly write the dependency on $\epsilon$ in Proposition 7; (ii) revise the statement to "Both proofs provide methods for building the corresponding networks, though not necessarily efficient in terms of depth or parameters"; and (iii) reformulate our expressivity claim to state that "FS-KANs and parameter-sharing MLPs are dense in the same function space, establishing equivalent expressive power."
> 4. **"The experimental results compare FS-KANs only with MLPs with shared parameters. Comparisons with other equivariant architectures should be considered."**
> Thank you for this comment. We focus our comparisons on parameter-sharing MLPs because they represent the standard and practical approach for building equivariant architectures. While alternative equivariant architectures  exist (e.g., models based on explicit group averaging like group convolutions and symmetrization-based approaches), these methods are computationally prohibitive for the symmetry groups we consider and are not used in practice for real-world applications. Parameter-sharing remains the dominant paradigm for equivariant learning precisely because it offers an efficient, scalable solution. Our comparisons, therefore, reflect the most relevant baselines that practitioners would actually deploy. Additionally, as established in our theoretical analysis (Propositions 6-7), FS-KANs and parameter-sharing MLPs have equivalent expressive power, meaning that comparing against parameter-sharing MLPs directly addresses questions about the fundamental capabilities of our approach relative to the practical state-of-the-art in equivariant architectures. If this does not fully address your question, we would appreciate clarification on which specific equivariant architectures you believe should be included, and we will make candid efforts to add them.

---

> > ### Author Response · Authors · 2025-11-20
> >
> > 5. **"Paragraph 'Invariance, equivariance, and standard parameter-sharing': The problem is formulated for $V = V' = \mathbb{R}^n$, instead of general $V = \mathbb{R}^n, V' = \mathbb{R}^{n'}$"**
> > Thank you for pointing this out. We appreciate the opportunity to clarify this notation. While our work indeed focuses on real-valued representations, we respectfully note that the suggested notation $V = \mathbb{R}^n, V' = \mathbb{R}^{n'}$ might be potentially confusing  in our context. As demonstrated throughout the paper, FS-KANs can operate on higher-order tensors (not just vectors), with group actions defined accordingly on these tensor spaces. The current notation $V = V' = \mathbb{R}^n$ in this particular section is used for simplifying the presentation of the core concepts, but the framework naturally extends to more general tensor product spaces as shown in subsequent sections and employed in our experiments. We will clarify this in the revision.
> >
> > 6. **"Equation (3) gives the wrong impression that only constant, zero-sum offsets are allowed. While this is clear from the context, I suggest modifying the example in (3) by using a functional offset."**
> > Thank you for this observation. We respectfully clarify that Equation (3) is intended as one illustrative example of an equivariant transformation, not an exhaustive characterization. However, the choice of constant, zero-sum offsets is not arbitrary: as demonstrated in our proofs, the equivariance constraint on KA layers fundamentally restricts the permissible form of offsets. Specifically, introducing functional (non-constant) offsets would violate the equivariance property of the layer. The constant zero-sum structure in Equation (3) is therefore a direct consequence of the equivariance requirement, not merely a simplification for presentation purposes.
> >
> > 7. **"Proof of Proposition 6: The notions of grid size and $k$-splines ('grid size $G = 2$ with degree $k$ B-splines') are nowhere defined before."**
> > We thank you for this note. We will add clarification in our revision. Here, $k$ determines the polynomial degree of both the activation and the spline basis functions, while $G$ specifies the number of grid intervals used in the spline parameterization.
> > We hope our responses have adequately addressed the reviewer's concerns and clarified the contributions of our work. Given the revisions we have made to strengthen the manuscript based on this feedback, we would be grateful if the reviewer would consider raising their score to a positive one.

---

> > > ### Comment · Reviewer_9HVo · 2025-11-26
> > > **Response to the Authors**
> > >
> > > I thank the authors for the detailed rebuttal and revisions. Several of my points, especially regarding the definition of the efficient FS-KA layers and the relation to prior equivariant KAN approaches, are now clearer, and the manuscript has improved.
> > > However, my main concerns about the novelty and the limited experimental scope remain only partially addressed. For this reasons I am keeping my score.

---

### Official Review · Reviewer_4Zk8 · 2025-11-02

**Soundness:** 3
**Presentation:** 3
**Contribution:** 3
**Rating:** 6
**Confidence:** 4

**Summary:**

The paper introduces FS-KAN, a principled framework for building permutation-equivariant and invariant KAN layers by sharing functions (rather than weights) across positions. The authors prove that FS-KANs have expressive power equivalent (on bounded domains) to parameter-sharing MLPs; this lets known universality and expressivity results for equivariant networks (e.g., DeepSets/GNN variants) transfer directly. Empirically, across point-cloud classification (ModelNet40, including a continual-learning protocol) and recommender datasets with $S_n \times S_m$ symmetry, FS-KANs show clear gains in low-data regimes and reduced forgetting, while preserving KAN interpretability. The paper also presents an “efficient” FS-KAN variant that reduces (but does not eliminate) runtime/memory overheads. Overall, the work unifies scattered equivariant-KAN designs and offers a clean blueprint to apply KA layers under arbitrary permutation groups.

**Strengths:**

- This paper presents an elegant generalization of parameter sharing to function sharing in KANs; the framework subsumes prior set/image KANs and handles arbitrary permutation groups and multi-channel tensors.

- The authors proves interesting bidirectional results: on the one hand, FS-KAN can be used to represent ant parameter-sharing MLPs; on the other hand, parameter-sharing MLPs can be used to approximate any FS-KANs. These results clarify exact expressivity and transfer known universality/limits from parameter-sharing MLPs to to FS-KANs.

- On low-data point-cloud and recommendation settings, FS-KANs deliver consistent improvements in accuracy/RMSE and lower forgetting in continual learning.

**Weaknesses:**

My main concern with this paper lies in the weaknesses of the experimental evaluation.

- Regarding compute/latency, even the efficient variant is ~1.5× faster than vanilla FS-KAN yet still slower than DeepSets, and its memory usage remains higher.

- Figure 5 and Table 1 show that FS-KAN’s improvements over the baselines are marginal; if anything, they suggest parity with older baselines such as Point Transformer.

- No large-scale benchmarks are discussed. Given the high computational overhead and only marginal gains over DeepSets, it is plausible that DeepSets may slightly surpass FS-KAN at larger training sizes.

However, because the theoretical results constitute the paper’s main contribution, and are both interesting and valuable to the ML community, I assign a positive score at this time.

**Questions:**

- What is the impact of the number/order of knots, the function parameterization (splines vs. alternatives), and the degree of sharing (full vs. partial) on performance and computational cost?

- How would the theoretical results change if the permutation group were replaced by a continuous group (e.g., rotations)?

- Is there an intuitive explanation for why FS-KANs can exactly represent any parameter-sharing MLP, whereas parameter-sharing MLPs can only approximate, but not exactly represent, arbitrary FS-KANs?

---

> ### Author Response · Authors · 2025-11-20
>
> We sincerely thank the reviewer for their positive evaluation and constructive feedback. We are pleased that the reviewer found our work to be **"an elegant generalization of parameter sharing to function sharing in KANs,"** appreciated our **"interesting bidirectional results,"** and recognized that **"on low-data point-cloud and recommendation settings, FS-KANs deliver consistent improvements"**. We are particularly encouraged by the reviewer's assessment that **"the theoretical results constitute the paper's main contribution, and are both interesting and valuable to the ML community."**  We address the reviewer's concerns below:
> 1. **"Efficient FS-KAN yet still slower than DeepSets"**
> We recognize that current KAN-based architectures present computational efficiency concerns; nonetheless, the broader KAN community is actively addressing these challenges through parallel efforts to both optimize existing implementations and explore novel application domains. A key strength of our approach is that the function-sharing mechanism we propose is independent of the underlying KAN parameterization—meaning that our architectural innovation will naturally inherit the benefits of any future efficiency improvements in KAN implementations without requiring redesign. We also emphasize that data-scarce scenarios are prevalent across numerous ML application areas. In these contexts, where data is limited, achieving better results at the cost of higher computational expense is often an acceptable and even preferable trade-off for practitioners.
>
>
> 2. **"No large-scale benchmarks are discussed"**
> To address the reviewer's concerns regarding scalability and large-scale benchmarks, we have included results from point cloud classification experiments in the high-data regime using **all** ModelNet40, a standard real-world benchmark in 3D computer vision with 8,859 training samples and 1,024 points per cloud.
>
> | Model          |   DeepSets    |    FS-KAN     | Efficient FS-KAN | Point-Transformer | Standard KAN  |
> |:---------------|:-------------:|:-------------:|:----------------:|:-----------------:|:-------------:|
> | Test Accuracy  | 0.844 ± 0.005 | 0.837 ± 0.007 | 0.840 ± 0.005    | 0.840 ± 0.008     | 0.232 ± 0.074 |
>
> This experiment complements our low-data regime results presented in the main paper. The results demonstrate that in the high-data regime, FS-KAN variants achieve comparable accuracy with established architectures like DeepSets and Point-Transformer. However, the significantly higher computational costs (both in terms of training time and memory usage) make KAN-based approaches less practical in high-data regimes, which is consistent with our findings in the signal classification task. We note that this aligns with our discussion in Section 5 and conclusion in Section 6, where we highlighted that FS-KAN models are particularly well-suited for low-data regimes, where their superior sample efficiency justifies the additional computational cost.
>
> 3. **"What is the impact of the number/order of knots, the function parameterization (splines vs. alternatives), and the degree of sharing (full vs. partial) on performance and computational cost?"**
> We have expanded the existing hyperparameter study (Appendix D) to include detailed tables comparing runtime and memory usage across different depths, widths, and grid sizes (the number of control points in the spline basis functions). These additional tables demonstrate how computational cost scales with each architectural parameter, complementing the existing accuracy analysis and providing practical guidance for model selection based on resource constraints. Our analysis reveals that depth 4 consistently achieves the best accuracy across configurations, while training time and memory usage scale super-linearly with width and more linearly with depth. Grid size shows moderate impact on both computational costs and accuracy, with larger values providing diminishing returns. Regarding the distinction between partial and full sharing mentioned in the review, we are not familiar with this terminology in the context of our function sharing architectures and would appreciate clarification from the reviewer to ensure we address this concern appropriately.

---

> ### Author Response · Authors · 2025-11-20
>
> 4. **"How would the theoretical results change if the permutation group were replaced by a continuous group (e.g., rotations)?"**
> Thank you for raising this point. Extending our framework to continuous groups would require fundamentally different technical approaches. While permutation groups allow for closed-form discrete sharing patterns, continuous groups do not admit such elegant solutions. Instead, they would typically require numerical methods to solve for layers that satisfy equivariant constraints, or use polynomial layers and other more complex techniques. Our focus on permutation symmetries is motivated by both practical and theoretical considerations: these symmetries enable efficient closed-form solutions and, importantly, appear across diverse data types—sets, graphs, point clouds, hierarchical structures, and numerous relational datasets encountered in scientific and industrial contexts. Exploring extensions to continuous groups would necessitate considerably different mathematical tools and falls outside the scope of our current work.
>
> 5. **"Is there an intuitive explanation for why FS-KANs can exactly represent any parameter-sharing MLP, whereas parameter-sharing MLPs can only approximate, but not exactly represent, arbitrary FS-KANs?"**
> Thank you for this insightful question. We emphasize that from a theoretical expressivity standpoint, the distinction between exact representation and approximation is not significant here: both Theorems 3.1 and 3.2 establish that FS-KANs and parameter-sharing MLPs are dense in the same function space, meaning both can approximate any target function arbitrarily well. The asymmetry in exact versus approximate representation is purely a technical artifact of our proof construction and does not reflect any fundamental difference in expressive power between the two architectures. Specifically, splines are piecewise polynomials, making the exact simulation of linear layers and ReLU activations (which are first-order piecewise polynomials) a straightforward task for KANs, while the reverse direction (approximating general splines with ReLU networks) requires approximation rather than exact representation.
>
> We sincerely appreciate the reviewer's positive evaluation of our work. We hope our detailed responses and the improvements made to the manuscript further strengthen the contribution, and we would be grateful if the reviewer would consider adjusting their score accordingly.

---

> > ### Comment · Reviewer_4Zk8 · 2025-11-27
> >
> > Thank you to the authors for the rebuttal. However, my concern about the marginal improvement, the slower performance compared to previous methods, and the lack of large-scale experiments still stands. I personally do not consider ModelNet40 a large-scale dataset. Therefore, I will keep my score as it is.

---

### Official Review · Reviewer_5F3P · 2025-11-02

**Soundness:** 2
**Presentation:** 3
**Contribution:** 2
**Rating:** 4
**Confidence:** 4

**Summary:**

The paper proposes a permutation-equivariant neural network based on the Kolmogorov–Arnold framework.

**Strengths:**

- The paper is well written and self-contained. Even for readers who are not deeply familiar with KANs, the main ideas are clear and accessible.

- The construction of the proposed permutation-equivariant and permutation-invariant KANs is clearly presented. It is supported by solid theoretical guarantees, including expressivity results (Propositions 2 and 4) and expressive power analysis (Section 4). The accompanying visualizations effectively illustrate the concepts, and to the best of my knowledge, this formulation is novel within the context of permutation-equivariant networks.

- The experimental section covers a diverse set of symmetry-structured tasks, such as signal classification, point-cloud classification, and semi-supervised rating prediction, demonstrating the versatility of the proposed framework.

**Weaknesses:**

- While the theoretical results are clearly presented and technically sound, they are somewhat expected and incremental. Propositions 2 and 4 follow naturally from basic algebraic reasoning, and the expressive power result largely derives from existing results on standard KANs.


- I find the motivation from the KAN literature somewhat confusing. KANs are often considered computationally heavy and inefficient compared to mainstream architectures such as MLPs, CNNs, or Transformers. Although spline-based parameterizations can alleviate some of these issues, KAN-based models are still known for their slower training and higher memory consumption. While the paper provides several theoretical guarantees on expressivity, I believe the empirical performance and efficiency aspects are more crucial in this context. **The trade-off between accuracy and computational cost remains too large**. This makes the practical necessity of applying KANs to permutation-equivariant networks questionable. Although the authors introduce an efficient variant of FS-KAN to mitigate these issues, the improvement is modest, and in some cases, the efficient version performs comparably or even worse than the full FS-KAN model.

**Questions:**

Could the authors provide additional evidence or comparisons regarding runtime and memory efficiency with other baselines? Currently, the analysis only includes DeepSets, making it difficult to assess how FS-KAN performs relative to a broader range of architectures.

---

> ### Author Response · Authors · 2025-11-20
>
> We sincerely thank the reviewer for their constructive feedback and thoughtful evaluation of our work. We are pleased that the reviewer found **"the main ideas are clear and accessible"** and that our work is **"supported by solid theoretical guarantees"**. We address the reviewer's concerns below:
>
> 1. **"The theoretical results are somewhat expected and incremental"**.
> We respectfully disagree with this assessment. While we build upon the established framework of weight sharing in equivariant neural networks, the extension to function sharing in KANs is non-trivial and has significant theoretical and practical implications. First, our theoretical results provide the first rigorous framework for incorporating symmetries into KAN architectures, addressing a gap in the literature where such analysis was previously absent. We recap and highlight the main contributions in the following:
>
> a. We illustrated how equivariant KA layers admit multiple representations (e.g., Equation 3), and established that every such layer has an equivalent function-sharing form (Propositions 2, 4). This required non-trivial proofs involving orbit decompositions.
>
> b. We extended the KAN framework to multiple feature channels and higher-order tensors (Proposition 5, Section 3.3), which required careful treatment. While our proofs leverage established algebraic tools from equivariant deep learning, this does not diminish their importance — rather, it demonstrates how these principles are to be carefully adapted to the function-based parameterization of KANs.
>
> c. We prove that FS-KANs and parameter-sharing MLPs have equivalent expressive power (Propositions 6-7). This equivalence is significant because it allows us to transfer well-established expressivity results from the equivariant MLP literature to FS-KANs (Corollary 4.1), including universality theorems for set functions and WL-test characterizations for graph networks. This connection provides both fundamental limitations and guarantees for FS-KAN expressivity that were not previously established.
>
> d. We provided extensive experimental validation across diverse data symmetries, something lacking in the existing KAN literature. The combination of theoretical foundations and comprehensive empirical study represents a meaningful contribution to understanding how KANs can exploit structural symmetries in data.
>
> 2. **"The trade-off between accuracy and computational cost remains too large"**.
> We appreciate the reviewer's concerns regarding computational efficiency. We acknowledge and clearly state in the paper that KAN-based models currently face computational challenges; however, improving the runtime of KANs is an active area of research in the *broader KAN literature*, with ongoing efforts to develop more efficient variants proceeding in parallel with explorations of KANs' applicability to new domains.<ins> Critically, our function-sharing framework will directly benefit from any future advances in KAN efficiency, as the architectural contribution, that is, the function-sharing mechanism for permutation equivariance, is orthogonal to the specific KAN parameterization used . *Any improvements in KAN implementations (faster spline implementations, alternative basis functions, etc.) can be seamlessly integrated into FS-KAN with minimal modification*. While we agree the current accuracy-efficiency trade-off may limit immediate widespread deployment, we view this work as fundamental research that establishes the viability of KAN-based permutation-equivariant architectures, which will become increasingly practical as the community develops more efficient KAN variants.
> We conclude this answer by also remarking the evidenced improved sample efficiency of our approach.  Low data setups are common in many fields that utilize ML tools, and we believe that attaining better results at a higher computational cost can still represent a compelling trade-off for practitioners.
>
> 3. **"Could the authors provide additional evidence or comparisons regarding runtime and memory efficiency with other baselines?"**
> Thank you for this suggestion. We provide comprehensive runtime and memory measurements for the signal classification task (Appendix C.2, Table 3) and the point cloud classification task (Appendix C.3, Tables 4 and 5), which we discuss in Section 5. In our revision, we have included measurements for Point Transformer and non-invariant KAN baselines as well, providing a more complete picture of computational trade-offs. If the reviewer has specific additional baselines in mind that would strengthen the comparison, we would be happy to make an effort to include their measurements as well.
>
>
> We hope our responses have adequately addressed the reviewer's concerns and clarified the contributions of our work. We would be grateful if the reviewer would consider raising their score to a positive one.

---

> > ### Comment · Reviewer_5F3P · 2025-11-26
> >
> > I thank the Authors for their response, and I provide several follow-up remarks below.
> >
> > - 1
> >
> > I maintain my previous view that many theoretical results, particularly those in Section 3.2, are somewhat expected. However, upon reconsideration of my earlier review, I acknowledge that these results remain necessary, as the proofs and tools involved may lie outside the standard background of a typical ML audience. Their inclusion does contribute to a clearer understanding of the framework.
> >
> > -  2+3
> >
> > I appreciate the additional clarification regarding the computational cost of the baselines, which makes comparison more transparent. I understand the heavy computation required by KAN architectures; ultimately, it should be the community’s decision whether such cost is acceptable for broader ML applications. One further question: I am surprised that Standard KAN performs considerably worse than FS-KAN. Could the Authors comment on why this disparity is so large?
> >
> > *Minor formatting suggestions (do not affect my score):*
> >
> > - All revised content in the manuscript should be clearly distinguished from the original text.
> >
> > - There are a few typographical issues, including some incorrect uses of \ref and \eqref (e.g., line 1072).
> >
> > - Consider removing vertical lines in tables for cleaner formatting (as venue policy).
> >
> > These are only cosmetic suggestions, but worthwhile to address given that revisions are allowed.
> >
> > ---
> >
> > Overall, despite the remaining open points requiring clarification, I find the contribution meaningful and potentially valuable to the community. I am raising my score to 6, with a positive attitude toward the submission.

---

### Comment · Area_Chair_aPaq · 2025-11-23
**Next Steps Following Authors’ Rebuttal: Review Rebuttal and Participate in Discussion**

Dear Reviewers,

Thank you very much for your thoughtful evaluations of this paper.

Now that the authors have submitted their rebuttal, I kindly ask you to take the following steps (if you have not done so already):

- Read the other reviews as well as the authors’ response.
- Consider whether the rebuttal and additional comments affect your assessment of the paper.
- Engage in interactive discussion with the authors **before November 25**, encouraging a dynamic exchange rather than a one-sided rebuttal.

The current reviews for this paper are mixed. Your contributions at this stage are essential for forming a well-informed final decision. I therefore ask that you reassess your views in light of the authors’ responses and the broader discussion among reviewers.

I am happy to join and support the discussions between you and the authors. Please feel free to share your thoughts and participate actively in the discussion.

Thank you once again for your service to ICLR 2026.

Best regards,

 AC

---

> ### Author Response · Authors · 2025-11-28
>
> We sincerely thank the AC for facilitating this discussion and for encouraging continued dialogue during the rebuttal period. We also deeply appreciate Reviewers 9HVo, 5F3P, G8oo and 4Zk8 for their constructive engagement and thoughtful responses to our rebuttal. We kindly ask the remaining reviewers to please address our responses to their concerns, as we would greatly value the opportunity for further discussion to ensure we have adequately addressed all questions. We remain committed to improving our work and welcome any additional clarifications or suggestions from all reviewers.
>
>
>
> Thank you again for your time and consideration.
>
> Best regards,
>
> Authors

---

### Author Response · Authors · 2025-12-03

We sincerely thank the new AC for taking on the review of our paper. To assist you in this process, we provide below a summary of the review outcomes and rebuttal discussions thus far.

The initial scores were 6, 6, 4, 4, 4, 2 from the six reviewers. Following the comprehensive rebuttal and discussion phase, scores shifted to 6, 6, 6, 6, 4, 2 (5F3P and G8oo raised their rating from 4 to 6) , with two reviewers (DxdS - score 2, and mRDw - score 6) not participating in the post-rebuttal discussion.

---

**Acknowledged strengths**

Reviewers jointly acknowledged several key strengths:

1. **Solid theoretical foundations:** Our theoretical framework was found to be "supported by solid theoretical guarantees" (5F3P), with results that are "both interesting and valuable to the ML community" (4Zk8) and represent "a rigorous result which requires new and different proofs from the ones known in the weight sharing literature" (DxdS).

2. **Strong empirical performance:** Reviewers noted that "..., FS-KANs deliver consistent improvements" (4Zk8) and that "empirical evidence suggests that their FS-KAN is better compared to other baselines" (G8oo).

3. **Clear presentation:** Multiple reviewers appreciated that "the main ideas are clear and accessible" (5F3P), the paper is "well written and clearly structured with working examples" (DxdS), and "the experimental section is clearly structured" (9HVo).

---

**Addressing reviewers’ points**

During the rebuttal period, we strengthened our submission by addressing reviewers' comments:

`. **Computational cost concerns:** questions about computational costs were raised by reviewers mRDw ("Despite the good performance, computational cost remains high") and 4Zk8 ("Efficient FS-KAN yet still slower than DeepSets"). We emphasized that our function-sharing framework is orthogonal to the specific KAN implementation and parameterization used, meaning it will directly benefit from any future advances in KAN efficiency being pursued by the broader research community. Any improvements in KAN implementations (faster spline methods, alternative basis functions, etc.) can be seamlessly integrated into FS-KAN, making our architectural contribution increasingly practical as KAN efficiency continues to improve.

2. **Enhanced computational analysis and efficiency comparisons:** Reviewers 5F3P ("Could the authors provide additional evidence or comparisons regarding runtime and memory efficiency with other baselines?"), 4Zk8, and mRDw requested more comprehensive efficiency comparisons with baselines. We included additional detailed runtime and memory measurements across different architectural configurations, while also adding more baselines to existing comparisons, providing practitioners with more concrete guidance.

3. **Large-scale evaluation**: Reviewers 4Zk8 ("No large-scale benchmarks are discussed") and mRDw ("The empirical analysis, while broad, relies mostly on synthetic or small-scale datasets") expressed concerns that experiments were limited to small-scale datasets. We conducted complementary experiments on the full ModelNet40 dataset (8,859 training samples), demonstrating that FS-KAN achieves comparable accuracy with established methods in high-data regimes while confirming its particular advantage in low-data settings.

4. **Improved theoretical clarity**: Reviewers 9HVo and DxdS requested additional clarifications about formulations and constructions in our theoretical results. We revised theoretical statements for greater formality and precision, clarified the construction of efficient FS-KAN layers for general permutation groups, and provided clearer exposition of technical concepts.

5. **Clarified scope and contributions**: Reviewers 5F3P ("The theoretical results are somewhat expected and incremental") and DxdS ("Insufficient novelty") raised concerns regarding the novelty of extending weight sharing to function sharing in KANs. We explained the non-trivial theoretical developments required for this adaptation, our framework's role in providing the first systematic treatment of symmetries in KAN architectures, and the practical significance of achieving superior performance in low-data regimes prevalent across scientific and industrial applications.

---

**Score updates and post-rebuttal reviewer feedback**

Following our responses, Reviewers 5F3P and G8oo raised their scores to positive ones, *resulting in a strong majority of reviewers (four out of six) with a positive rating of 6.* We believe our comprehensive responses and additional experimental results have strengthened the paper substantially and addressed the core concerns raised during the rebuttal process. We remain available to provide any additional clarifications or materials that would assist in your evaluation.

Best regards,

The Authors

---

### Meta-Review · Area_Chair_JMLK · 2026-01-07

**Summary:**

The paper introduces permutation equivariance to Kolmogorov-Arnold networks, and proves several results about its generality and expressive power.

Initial reviews were mixed. Reviewers praised the good writing, solid and elegant theory, and model performance in low-data regimens. Several concerns were raised about high computational cost, unconvincing experiments, and lack of novelty.

The rebuttal partially addressed those concerns. The method seems to have very little practical application at this point, although it might benefit from future improvements to KAN. The theory is solid and new, but some saw it as incremental.

Overall this is a borderline case. However, during the shortened discussion period, two reviewers indicated that they would increase their scores from 4 to 6 so the outlook is positive. I recommend acceptance, despite the lack of reviewer consensus.

**Reviewer Concerns:**

Almost all reviewer raised concerns about the high computational cost and unconvincing experiments. The rebuttal clarified that the high computational cost is inherent to KANs and orthogonal to the contributions. The rebuttal emphasized that most advantages of the method are in low-data, while providing a slightly larger experiments on ModelNet40 that shows the proposed method matching the baselines. In my view these concerns were only partially addressed; ModelNet40 is still synthetic and quite small. The method seems to have very little practical utility at this point, although it might benefit from future improvements to KAN.

5F3P and DxdS made a point that the theoretical contributions are small and follow directly from prior work. The rebuttal argued that the theoretical results have not appeared before and reiterated the contributions, which seemed to satisfy 5F3P but DxdS did not respond. This point was only partially addressed.

**Reviewer Scores:**

5F3P's concerns were partially resolved and they indicated willingness to increase the score during discussion (4 to 6).

4Zk8, mRDw had initial positive recommendations and their concerns were partially resolved; they would likely keep the score.

9HVo was not satisfied with the novelty and experimental evaluation after the rebuttal so they would not change the negative recommendation.

DxdS's concerns about lack of novelty were partially addressed, they would likely increase their score but likely to a 4 and not 6.

G8oo raised somewhat minor experimental weaknesses which were addressed; the reviewer responded positively and would increase the score (4 to 6).

---

### Decision · Program_Chairs · 2026-01-26

Accept (Poster)